# Aerosol measurements during COPE: composition, size and sources of CCN and INPs at the interface between marine and terrestrial influences

J. W. Taylor[1], T. W. Choularton[1], A. M. Blyth[2], M. J. Flynn[1], P. I. Williams[1,3], G. Young[1], K. N. Bower[1], J. Crosier[1,3], M. W. Gallagher[1], J. R. Dorsey[1,3], Z. Liu[1], P. D. Rosenberg[4]

[1] Centre for Atmospheric Science, School of Earth, Atmospheric and Environmental Sciences, University of Manchester, Manchester, UK

[2] National Centre for Atmospheric Science, University of Leeds, Leeds, UK

[3] National Centre for Atmospheric Science, University of Manchester, Manchester, UK

[4] School of Earth and Environment, University of Leeds, Leeds, UK

*Correspondence to*: Jonathan W. Taylor (jonathan.taylor@manchester.ac.uk)

**Abstract.** Heavy rainfall from convective clouds can lead to devastating flash flooding, and observations of aerosols and clouds are required to improve cloud parameterisations used in precipitation forecasts. We present measurements of boundary layer aerosol concentration, size and composition from a series of research flights performed over the southwest peninsula of the UK during the COnvective Precipitation Experiment (COPE) of summer 2013. We place emphasis on periods of southwesterly winds, which locally are most conducive to convective cloud formation, when marine air from the Atlantic reached the peninsula. Accumulation mode mass loadings were typically $2 - 3$ µg m$^{-3}$ (corrected to standard m$^3$ at 1013.25 hPa and 273.15 K), the majority of which was sulphuric acid over the sea, or ammonium sulphate inland, as terrestrial ammonia sources neutralised the aerosol. The cloud condensation nuclei (CCN) concentrations in these conditions were ~$150 - 280$ cm$^{-3}$ at 0.1% and $400 - 500$ cm$^{-3}$ at 0.9% supersaturation (SST), which are in good agreement with previous Atlantic measurements, and the cloud drop concentrations at cloud base ranged from $100 - 500$ cm$^{-3}$. The concentration of CCN at 0.1% SST was well correlated with non-sea-salt sulphate, meaning marine sulphate formation was likely the main source of CCN. Marine organic aerosol (OA) had a similar mass spectrum to previous measurements of sea spray OA, and was poorly correlated with CCN.

In one case study that was significantly different to the rest, polluted anthropogenic emissions from the southern and central UK advected to the peninsula, with significant enhancements of OA, ammonium nitrate and sulphate, and black carbon. The CCN concentrations here were around six times higher than in the clean cases, and the cloud drop number concentrations were $3 - 4$ times higher.

Sources of ice-nucleating particles (INPs) were assessed by comparing different parameterisations used to predict INP concentrations, using measured aerosol concentrations as input. The parameterisations based on total aerosol produced INP concentrations agreed within an order of magnitude with measured first ice concentrations at cloud temperatures as low as -12°C. Composition-specific parameterisations for mineral dust, fluorescent particles and sea spray OA were $3 - 4$ orders of

magnitude lower than the measured first ice concentrations, meaning either a source of INPs was present that was not characterised by our measurements, and/or one or more of the composition-specific parameterisations greatly underestimated INPs in this environment.

## 1 Introduction

Flash flooding occurs when rain is heavy and persistent. The southwest peninsula of the UK (shown in Figs. 1 and 2) is particularly prone to flash flooding because convective clouds form along convergence lines that often persist for many hours, as was the case in the Boscastle flash flood of August 2004 (Golding et al., 2005). This type of event develops when the wind blows from the southwest and aligns with the peninsula. Convergence associated with sea breeze fronts from the north and south coasts generates quasi-stationary convective systems (Warren et al., 2014). In the Boscastle case, individual

cells moved with the wind, but new cells were constantly generated at the upwind end, causing the system as a whole to remain in a quasi-stationary position. Each cell developed over a similar timeframe, and they therefore precipitated over the same region, causing highly localised flooding. The potential of convective clouds to generate persistent, heavy rainfall, and consequent flooding, along the peninsula demonstrates the importance of understanding cloud formation and development in the region.

As convection is initiated in the newly developing cells, boundary layer aerosols are lifted up to provide the cloud condensation nuclei (CCN) necessary for cloud drop formation. Increased concentrations of larger and more soluble aerosols enhance CCN numbers and, consequently, enhanced cloud drop number concentration (CDNC). This inhibits the formation of warm precipitation by reducing average drop size (Fan et al., 2007; Huang et al., 2008), which also affects riming efficiency in mixed-phase clouds (Klett and Davis, 1973). At sub-zero temperatures, aerosols provide the ice nucleating

particles (INPs) required for primary ice formation, which may then lead to secondary ice processes such as the Hallett-Mossop rime-splintering processes(e.g. Huang et al., 2008; Phillips et al., 2001). Interactions between the warm rain and secondary ice processes are thought to be key to determining the timing and location of cold precipitation (Taylor et al., 2016). Aerosols may influence these processes by affecting cloud and drizzle drop size, and by determining the concentrations of primary ice forming at different cloud temperatures. Koren et al. (2005) found that convective clouds

forming in polluted air may have a higher cloud fraction, lower ice fraction and lower droplet radius than those forming in clean Atlantic air.

In the prevailing west/southwesterly winds, Atlantic air influences the southwest peninsula, whereas in northerly or easterly winds the region receives anthropogenic pollution from the UK and Western Europe. Local emissions from farming and transport may also influence the aerosol over the peninsula.

The COnvective Precipitation Experiment (COPE) (Leon et al., 2015) took place in the southwest peninsula during July-August 2013. Multiple aircraft made in-situ measurements of aerosol and clouds, as well as airborne remote sensing

measurements, on days predicted to be favourable for convective cloud formation. Additional ground-based aerosol and X-band radar measurements were performed throughout the duration of the experiment. The main aim of COPE was to improve quantitative precipitation forecasting in numerical weather prediction (NWP) models. Clark et al. (2016) describe state-of-the-art convection-permitting models, which are used operationally for rainfall forecasting both in the UK and

elsewhere. The development and application of these NWP models requires the use of parameterisations for processes that would be too small in scale or too costly computationally to explicitly evaluate on an operational basis. A parameterisation may be used, for example, to calculate the autoconversion rate of cloud water to rain, rather than explicitly evaluating processes such as collision-coalescence, condensation and evaporation over sub-grid timescales.

The development of improved parameterisations, and investigating the relative importance of different processes, involves

the use of more detailed research models. For example, Connolly et al. (2009) described the aerosol-cloud-precipitation interaction model (ACPIM), one of several research models which investigators will use as part of COPE to study different cloud-aerosol interactions and microphysical processes. ACPIM requires information on aerosol composition and size distribution, as well as environmental variables, to initialise its explicit cloud microphysics scheme. The purpose of this paper is to describe the boundary layer aerosol measured during the aircraft campaign in terms of concentration, size, and

composition, and to determine the main sources of CCN and INPs, in order to inform modelling studies aiming to improve forecasts of convective precipitation in the region. Additional measurements of cloud microphysics (e.g. Taylor et al., 2016) may be used to evaluate the importance of different cloud microphysical processes, while the X-band radar measurements provide a dataset to evaluate the accuracy of the distribution and rate of precipitation in a regional model.

## 2. Experimental

### 2.1 Aircraft measurements

Most of the data used in this analysis were measured by instrumentation mounted on the UK BAe-146-301 Atmospheric Research Aircraft during six case study flights over the southwest peninsula, which took place between 05 July and 03 August 2013. Each flight had a designated flight number, which are listed in Table 1. Additional airborne measurements were performed with the University of Wyoming King Air 200T (UWKA) and UK Met Office Civil Contingency 208

Aircraft (MOCCA), but these are not discussed here. The BAe-146 was fitted with a variety of aerosol and cloud probes. Here we only discuss instruments relevant to our analysis; Leon et al (2015) provide a full list of instrumentation aboard the aircraft. All concentrations are corrected to standard temperature (273.15 K) and pressure (1013.25 hPa). A typical flight plan involved performing below-cloud aerosol runs in the boundary layer before making measurements of clouds at higher levels. The flight plans of the boundary-layer aerosol runs in each flight are shown in Figure 2, and the altitude and duration

of the aerosol runs are listed in Table 1. Vertical profiles of potential temperature showed the boundary layer height was ~

750 – 1250 m above mean sea level (AMSL; unless otherwise stated, all altitudes henceforth are AMSL), meaning these aerosol runs were conducted in the boundary layer.

The concentration of condensation nuclei (CN, i.e. the aerosol number concentration) larger than 2.5 nm was measured using a model 3786-LP water-filled condensation particle counter (WCPC; Hering et al., 2005) manufactured by Aerosol Dynamics Inc., based on the TSI model 3786 but modified for use at low pressure. The accuracy of the CN concentration measurement is ±12% .

A Droplet Measurement Technologies (DMT) dual-column cloud condensation nuclei counter (CCNC; Roberts and Nenes, 2005) measured the concentrations of CCN at nominal supersaturations (SST) of 0.1% and 0.9%, though on 18 July the higher SST was set to 0.6%. The relative uncertainty in the supersaturation measurements is ±10% (Roberts et al., 2010; Trembath, 2013), and the uncertainty associated with the flow calibration and counting efficiency of the particle counter is typically ~6% (Trembath, 2013, p189). The CCNC sampled behind a pressure-controlled inlet set to 650 hPa.

The aerosol size distribution was measured with a custom-built inboard scanning mobility particle sizer (SMPS; Wang and Flagan, 1990), and wing-mounted Particle Measurement Systems passive cavity aerosol spectrometer probe (PCASP) - 100X, with electronics upgraded by DMT, and DMT cloud droplet probe (CDP). The PCASP and CDP were size calibrated as described by Rosenberg et al. (2012), and we assumed a refractive index of $n = (1.5 - 0i)$. The SMPS measures the mobility diameter of particles 20 – 320 nm, while the PCASP and CDP measure the optical diameter of particles 0.12 – 3.4 μm and 2 – 50 μm respectively. The CDP measures aerosols and cloud drops at ambient humidity. The PCASP inlet has a dried sheath flow, and the SMPS inlet line was dried to <45% RH by ram heating and the temperature increase as the sample line entered the cabin. The SMPS data were normalised to the PCASP by scaling the SMPS concentrations in each bin so the concentrations in the overlap region matched those of the PCASP. As is shown in later analysis, the particles are composed predominantly of secondary aerosol and are likely to be quasi-spherical, meaning any shape correction for the SMPS distributions would be minimal. The difference in the size of the PCASP bins derived using refractive index of 1.5 compared to using values 1.53 for ammonium sulphate and 1.44 for sulphuric acid is ~5%, which is equivalent to half a bin. Taking this difference into account when normalising the SMPS distributions gave a 3 – 6% uncertainty in the normalised SMPS distributions.

The composition of nonrefractory submicron aerosol was measured with an Aerodyne Research Inc. compact time-of-flight aerosol mass spectrometer (AMS; Canagaratna et al., 2007; Drewnick et al., 2005), which reports concentrations of organic aerosol (OA), sulphate ($SO_4^{2-}$), nitrate ($NO_3^-$) and ammonium ($NH_4^+$) as standard. Additionally, we used the fragmentation table given by Langley et al (2010) to calculate concentrations of methanesulphonic acid (MSA), and adjusted the fragmentation tables of OA and $SO_4$ accordingly. The AMS can also be used to report chloride ($Cl^-$) concentrations, which would be likely to be influenced by sea salt in a marine environment. Previous studies have attempted to quantify sea salt using the AMS, but the scaling factor used to correct the data for the instrument's poor detection efficiency of sea salt was highly variable (Nuaaman et al., 2015; Ovadnevaite et al., 2012; Schmale et al., 2013; Zorn, 2009). As no calibrations for sea

salt were carried out in the field, we are unable to estimate the scaling factor, and thus the Cl$^-$ concentration, with any reasonable accuracy, and we therefore do not report Cl$^-$ concentrations in this analysis. The SO$_4^{2-}$ measurement may have a contribution from sea-salt sulphate, but this is likely to be of the order of 1% (Schmale et al., 2013).

The AMS was calibrated at the start and end of each flight using monodisperse ammonium nitrate to calculate the ionisation
efficiency (IE) for NO$_3^-$, corrected as described by Morgan et al. (2014, Supplementary material). The ammonium nitrate calibrations were also used to calculate the relative ionisation efficiency (RIE) of NH$_4^+$, and a separate ammonium sulphate calibration was performed to calculate the RIE of SO$_4^{2-}$. The AMS IE calibration was stable within ±12% throughout the campaign, and the absolute accuracy is ~30% (Bahreini et al., 2009). The collection efficiency was calculated as recommended by Middlebrook et al. (2012), and the aerosol volume, calculated with volume mixing using organic and
inorganic densities of 1.27 g cm$^{-3}$ and 1.77 g cm$^{-3}$ respectively (Cross et al., 2007), showed qualitative agreement with the aerosol volume measured by the SMPS and PCASP.

Refractory black carbon (BC) concentrations were measured with a DMT single-particle soot photometer (SP2; Schwarz et al., 2010), which is the same instrument used in previous studies on the BAe-146 (e.g. Taylor et al., 2014), but upgraded to be functionally identical to the current SP2 model D. SP2 calibrations were carried out at the start of each flight, and in the
laboratory after the campaign. The incandescence detectors, which measure BC mass on a single-particle basis, were calibrated using 300 nm mobility diameter Aquadag, corrected for the difference in response between the calibration particles and ambient soot (Baumgardner et al., 2012; Laborde et al., 2012b). Using this calibration, the uncertainty in the reported BC mass concentrations are ±10% when sampling diesel soot, and ~20% when sampling biomass burning (Laborde et al., 2012a, 2012b).

The online inboard aerosol instrumentation (i.e. the AMS, SP2, SMPS and CCNC) sampled using Rosemount inlets, which have sample efficiencies of approximately unity for particles smaller than 600 nm in diameter when sampling marine aerosol (Trembath et al., 2012).

Bulk aerosol was collected on two 47-mm Nuclepore filters, with 10µm and 1µm pore sizes, connected in serial. The filter sampling system is described by (Formenti et al., 2008). The size and composition of particles collected on the filters was
determined offline after the campaign using scanning electron microscopy (Phillips FEI XL30 Environmental Scanning Electron Microscope with Field-Emission Gun (ESEM-FEG) , with an Energy-Dispersive X-Ray Spectroscopy (EDX) system (Hand et al., 2010)). Particles were classified by composition using the scheme described by Young et al. (2016). An example filter composition measurement from the COPE campaign is presented by Leon et al. (2015). In our analysis, we only use the filter measurements to estimate the concentration of mineral dust for use in INP parameterisations.

The CO concentration was measured using an AeroLaser VUV fluorescence monitor model AL5002 (Gerbig et al., 1999), and the 3-D wind vector was measured with a de-iced Aventech aircraft-integrated meteorological measurement system (AIMMS)-20 turbulence probe (Beswick et al., 2008). A Stratton Park Engineering Company SPEC 2DS stereo probe was used to derive ice concentration, using the data processing rules described by Taylor et al. (2016).

## 2.2 Ground site measurements

The COPE ground site (located at 50.6466°N, 4.63323°W) hosted a suite of aerosol and meteorological instrumentation operating continuously throughout July and August 2013. Leon et al. (2015) provide a full list of instrumentation at the ground site. The aerosol properties measured at the ground site followed the same trends as those measured aboard the BAe-146, though it is not in the scope of this paper to provide an in-depth comparison. In this analysis, we only use the size-resolved measurements of fluorescent aerosol made by a University of Hertfordshire wideband integrated bioaerosol sensor (WIBS) 4M, as this measurement was not available on the BAe-146. The WIBS was mounted atop the ~8 m-high sampling tower. The particle size distribution in the range 0.4 – 12 μm (optical equivalent diameter) is measured by light scattering at a wavelength of 635nm. When a scattering particle is detected, two xenon lamps fire sequential UV pulses at 280 and 370 nm, which can cause fluorescent emissions that are detected in three channels measuring the ranges 310–400 nm (channel F1), 420–650 nm (channel F2), and 420–650 nm (channel F3) (Healy et al., 2012). We consider the fluorescent particles measured by the WIBS to be primary biological aerosol particles (PBAP), Although the WIBS may also detect other fluorescent particles, the fluorescent particle concentration is generally considered to be a lower limit on the PBAP concentration (Huffman et al., 2010).

## 3. Results

### 3.1 Airmass history

Figure 1 shows 5-day back trajectories for each of the case study flights, calculated using the Hybrid Single Particle Lagrangian Integrated Trajectory Model (HYSPLIT) (Draxler and Hess, 1998). Trajectories were initiated every 60 seconds from the aircraft flight track during the boundary layer aerosol runs. The trajectories were performed with 3-hourly 0.5° Global Data Assimilation System (GDAS) reanalysis data, using full vertical dynamics on 23 model levels. Two additional trajectories were released from the ground site (at an altitude of 10m above ground level) on 03 August at 1100 and 1200UTC, the same time period as the aerosol runs, as the ground site measurements are used on this date in Sect. 3.5. These two ground-site trajectories follow a similar path to the ones released from the aircraft flight track on 03 August, meaning the aircraft and ground site were sampling from the same airmass on this date. Although the turbulent mixing in the boundary layer means the accuracy of individual trajectories is uncertain, examining the general trend provides information on the history of the airmass, and possible changes in its cloud nucleating potential, which is determined by synoptic-scale winds. Figure 2 shows the flight path the aircraft took during the below-cloud runs, and the boundary layer wind speed and direction measured during these runs. The wind data allow us to assess the accuracy and potential influence of the back trajectories' final stages, and examine local wind fields such as sea breeze circulations that are not captured by the meteorology data used to initiate the trajectories.

Figures 1 and 2 panels (c), (e) and (f) show that the 25 and 29 July and 03 August case studies shared the synoptic features of the "Boscastle" type meteorology, i.e. winds blowing marine air from the North Atlantic over the peninsula from the southwest (hereafter referred to as southwesterly winds (SWW)) (Golding, 2005). The back trajectories showed that the airmasses had been over the North Atlantic for at least five days prior to landfall. They also showed sharp changes in

direction over the ocean; comparison with UK Met Office synoptic charts (available at http://www1.wetter3.de/error_adblocker_archiv_ukmet.html) showed these were due to circulation around low-pressure systems as they moved east, and were associated with frontal precipitation. This precipitation is likely to have washed out the majority of any long-range transported aerosol as the trajectories show the airmass remained in the lower troposphere, below the likely source of frontal precipitation. On 28 July, the situation was reasonably similar in that the airmass had spent

time over the North Atlantic, but reached the peninsula from the south, and may have passed over the northwest tips of Spain and France. Local sea breeze circulations may still have caused convergence on these four days, but they had only a minor influence on the absolute wind speed and direction, as the synoptic-scale winds were so strong.

The meteorology on 05 and 18 July were markedly different to the other case studies. On these two days, high pressure systems located over the UK meant that synoptic-scale boundary layer winds were much weaker. The measured winds were

generally only a few metres per second and the greater variation in wind direction suggest that the local sea-breeze circulations had more of an influence on absolute wind direction on these days. On 05 July, the back trajectories show that for the few days prior to being sampled, the airmass had passed over the Atlantic, though the easternmost trajectories passed over the southern tip of Ireland and the south coast of Wales. It is difficult to assess the accuracy of these parts of the trajectories when local winds were dominant, though there was some anthropogenic influence on this flight, as discussed in

the next section. The trajectories for 18 July suggest the air over the peninsula had passed over many parts of England, Wales and Ireland, and are therefore likely to have also had a strong anthropogenic influence.

The colours in Fig. 2 divide sections along the flight track into "Northwest (NW) coast", "Inland" and "Other" influenced regions. The "NW Coast" regions include areas over the sea, and the regions labelled "Other" are not used in this analysis. Depending on the wind directions, aerosol in these regions may be influenced by marine or terrestrial emissions, or both. The

trajectories and wind data in Figs. 1 and 2 show that for flights on 05, 25 and 29 July, and 03 August, the sections on the NW coast were affected by marine air that had not been influenced by land, though this was not the case for the other two flights. Dividing the data into these sections allows us to investigate changes in aerosol chemistry and properties at the land-sea interface. This topic is discussed further in the next section.

### 3.2 Aerosol size distributions

Figure 3 shows the submicron aerosol size distributions measured during the six case study flights. The standard deviations are plotted to show the spatial variation over the runs, and were much larger than the calculated standard errors for the size range shown. Three submicron aerosol modes are evident from the distributions: nucleation, Aitken, and accumulation

modes, with varying magnitudes in the different flights. Coarse-mode aerosols were also measured, but the supermicron distribution is not shown in Fig. 3 as the bin sizing errors and overlap between the PCASP and CDP made the features difficult to distinguish. The mean diameters of the Aitken, accumulation and coarse modes are listed in Table 2. The mean diameter of the nucleation modes was likely to be smaller than the SMPS can reliably measure.

Many of the measured distributions did not drop to near zero in the lowest bin, suggesting nucleation mode aerosols were present smaller than the 20nm size of the smallest bin. In all flights, the concentrations of particles in all bins <40 nm in diameter was higher inland than over the sea. Additionally, the total CN concentration (listed in Table 1) was always higher and more variable inland, and the standard deviations of smallest particles measured were higher inland. These measurements suggest there was a source of small particles forming inland, which was not uniform in spatial location.

On most flights, the Aitken mode was centred on 40 – 60 nm, and the accumulation mode was centred around 115 – 190 nm. The gap in between was characteristic of the "Hoppel dip", thought to be caused by cloud processing of marine aerosol (Hoppel et al., 1986). Similar size distributions have been measured in summertime North Atlantic aerosol (Asmi et al., 2011; Dall'Osto et al., 2011). The accumulation mode on 05 July was larger, with modal diameter ~275 nm over the sea, which may be due to increased photochemical aerosol formation in the clear weather. 18 July was also an exception,

showing just one wide accumulation mode both over the NW coast and inland, and the size distribution was consistent with aged UK urban aerosol (e.g. Bohnenstengel et al., 2015) with the airmass having come from the east. The features of the marine aerosol size distribution seen on other days, such as a prominent Aiken mode and Hoppel dip, were also absent on this day due to the lack of a marine influence.

    It is also of interest to examine the concentrations of giant (dry diameter, $1 \leq D_p \leq 10$ μm) and ultragiant particles ($D_p > 10$

μm), as these may have an enhanced effect on precipitation formation via the warm rain process (Johnson, 1982), and our observations may be used to inform modelling studies (e.g. Blyth et al., 2013). The number concentrations of giant ($N_{GA}$) and ultragiant ($N_{UGA}$) are listed in Table 1. $N_{GA}$ varied by around a factor of two between cases, with values of around one per cubic centimetre, but there was no obvious trend to explain the differences between the different days. For the SWW cases, the inland $N_{GA}$ was fairly proportional to the NW coastal values, but lower by 15 – 40%. These are likely to be sea spray

aerosols, and indeed sea salt made up nearly 80% of supermicron aerosol volume on 3$^{rd}$ August (Leon et al., 2015). Additionally, on these days the mean diameter of the coarse aerosol was larger over the NW coast than inland. These measurements are consistent with dilution/deposition of sea salt particles inland being greater than any terrestrial coarse aerosol emissions such as PBAP or dust on these days.

    The concentrations of ultragiant aerosol also showed around a factor of two variation between the different cases, and were 3

– 6 L$^{-1}$, other than the NW coastal measurement on 29 July, which was 11 L$^{-1}$. Again, it is difficult to pick out any clear trend in the values. On some days, $N_{UGA}$ was higher over the sea, whereas on others it was higher inland. In all cases, $N_{GA}$ and $N_{UGA}$ comprised a very small fraction of the total aerosol number concentration.

### 3.3 Submicron aerosol composition

#### 3.3.1 Marine aerosol

Figure 4 shows the average aerosol composition measured by the AMS and SP2 in the NW coastal regions during the six case study flights. The data from Figure 4 are also listed in Table 1. In clean Atlantic air (NW coast on 05, 25 and 29 July and 03 August, as identified in the previous section) the composition was fairly consistent. The aerosol acidity, investigated here in terms of sulphate neutralisation, influences gas-particle partitioning. For example, nitrate aerosol cannot stably coexist when internally mixed with particle-phase sulphuric acid, but may do so with ammonium sulphate. Around $60 - 70\%$ of submicron aerosol mass was sulphate, in the form of a mixture of sulphuric acid and ammonium bisulphate as there was insufficient ammonium for full neutralisation. This is in agreement with measurements of North Atlantic aerosol made by Ovadnevaite et al. (2014).

In the marine airmasses, the ratio of $MSA/SO_4$ was ~0.05, which is perhaps lower than average, but within the range typically observed in clean marine air in summer (e.g. Dall'Osto et al., 2010; Huebert et al., 1996; Ovadnevaite et al., 2014; Phinney et al., 2006). MSA and sulphate were well correlated ($R^2 = 0.66$), and the source of both is likely to be oxidation of biogenic dimethyl sulphide (DMS) emissions. OA was the only other major aerosol component, though the ratio of $SO_4/OA$ did show some variation. It is unlikely that anthropogenic pollution, either from shipping or long-range transported, made any significant contribution to the aerosol loadings; in clean Atlantic air, the OA showed no correlation with BC ($R^2 = 0.0$), while $SO_4$ showed a weak negative correlation ($R^2 = 0.15$). These results are in agreement with a similar assessment of Atlantic aerosol made at Mace Head, Ireland (O'Dowd et al., 2014). The source of the marine OA is discussed further in Sect. 3.2.3.

The total mass loading measured by the AMS + SP2 varied from $2.4 - 3.8$ µg m$^{-3}$ in these flights. Black carbon was only a minor component and low, but non-zero levels of nitrate were measured, despite the fact that aerosol nitrate is not expected to form in acidic conditions as the sulphate displaces any nitrate to the gas phase. Interpretation of this apparent measured nitrate is discussed in Sect. 3.2.4.

#### 3.3.2 Inland aerosol

Submicron inland aerosol composition on flying days is shown in Figure 5. On flying days between $25 - 29$ July and 03 August, inland aerosol had composition similar to the Atlantic marine air, but with the addition of ammonium. On 25 and 28 July, the ammonium was sufficient to fully neutralise the sulphate, but on 29 July and 03 August, the sulphate was a mixture of ammonium sulphate and bisulphate. The total measured mass loadings were within 10% of the marine air for the corresponding flight.

Compared to the last four flights, the inland aerosol on 05 July had a much larger OA fraction. The low wind speeds on this day mean local aerosol sources are likely to have had more of an effect. Inland on 05 July, OA had a correlation with BC of

$R^2 = 0.56$, and with the aircraft's CO measurement an $R^2 = 0.69$. Additionally, the BC mass loading was around a factor of 10 higher inland than in the Atlantic air. Taken together, this suggests that local fossil fuel emissions were at least partly responsible for the increased OA inland on 05 July. Some of the easternmost trajectories passed over the southern tips of Wales and Ireland, which are an alternative source of the increased anthropogenic influence inland on 05 July.

5  18 July was somewhat of an outlier in terms of the COPE case studies in that the aerosol mass loadings were much higher than the other flying days. The back trajectories showed that the airmass had come from England, Wales and/or Ireland, so a strong anthropogenic influence would be expected, in contrast to the clean marine air from the southwest seen in the other cases. There was not much difference between the aerosol measured inland and on the NW coast, as the air over both areas had come from the East, bringing anthropogenic pollution to the region. OA was the largest component, followed by similar 10  amounts of ammonium nitrate and sulphate, which were both fully neutralised by ammonium.

### 3.2.3 OA composition and origin

Examining the ratios of different fragments in the OA mass spectrum can give an indication of OA sources as well as their level of ageing. AMS factor analysis most-commonly tends to pick out factors with differing levels of oxidation. Fresh fossil fuel emissions appear as hydrocarbon-like OA (HOA), while more aged aerosol including secondary OA (SOA) appears as 15  oxygenated OA (OOA), which may be semi-volatile (SV-OOA) or more aged low-volatility (LV-OOA) (Jimenez et al., 2009).

OA measured in marine background air may be primary OA from sea spray (Ovadnevaite et al., 2011), SOA from marine VOC emissions (Decesari et al., 2011), or long-range transported pollution. Globally, primary marine OA is thought to dominate over marine SOA formation (Arnold et al., 2009; Gantt and Meskhidze, 2013). As discussed in Sect. 3.2.2, local 20  anthropogenic emissions contribute to OA concentrations inland, while the marine OA would have less of an influence in areas further away from the sea.

The OA mass spectrum measured over the NW coast on 05 July is shown in Fig. 6. The spectrum is dominated by peaks at m/z 28 and 44 from $CO^+$ and $CO_2^+$, which are prescribed to be equal in the default fragmentation table. As is typical in oxidised OA spectra, these peaks are much larger than m/z 43 from $C_3H_7^+$ or $H_3C_2O^+$, as the $CO^+$ and $CO_2^+$ ions are more 25  likely to be formed by fragmentation of more oxidised organic molecules. Table 3 presents correlations between NW coast OA spectra from 05, 25 and 29 July, and 03 August, with several spectra from literature. The literature spectra are HOA and two OOA factors (Morgan et al., 2010), sea spray (Ovadnevaite et al., 2011), marine biogenic SOA (Chang et al., 2011, which was well correlated with SO4), and the "Organic factor" measured in the Arctic ocean by Chang et al. (2011), which was thought most likely to be sea spray OA and was poorly correlated with sulphate. Though there are many previous 30  measurements of different OA components in Europe, we choose those from Morgan et al (2010) because they were measured with the same instrument aboard the BAe-146, so any inter-instrument variability should be minimised. OOA-1 was more oxidised than OOA-2, but no assessment was made of their volatility. To ensure the comparison was fair, only m/z

values where all spectra had data points were used. The marine OA spectra showed the best correlation with sea spray OA, indicating this is a likely source, though the correlations with the other oxidised factors were still reasonably good.

Figure 7 shows the fractions $f_{44}$ and $f_{43}$ (the fraction of OA measured at m/z 44 and 43 respectively) for all six case study flights, divided up into NW coastal and inland data. The literature values from Table 3 are also plotted. Ambient OOA measurements tend to fall within the diagonal dashed lines in Fig. 7 (Ng et al., 2010), and fossil fuel emissions may be expected to fall close to the HOA point from Morgan et al. (2010) upon emission and progress approximately through OOA-2 to OOA-1 with increased oxidation.

The four marine spectra from this study have $f_{44}$ indicative of a moderate level of oxidation but, on average, slightly low $f_{43}$. Of the selected literature values, the marine OA all fall closest to the sea spray OA, rather than the marine biogenic. If the inland OA was composed of a mix of marine and terrestrial OA, lines drawn from the marine data points through their corresponding inland points should intersect in a region representative of terrestrial OA. For the four flights with clean marine spectra, the lines would intersect in a region around (0.1, 0.1), close to the SV-OOA data point from Morgan et al (2010). This is consistent with the terrestrial aerosol source being similar on each flying day, and being moderately oxidised.

It is not fully clear what fraction of the terrestrial OA is anthropogenic or biogenic. It is difficult to pick out anthropogenic locally produced SOA from biogenic as the mass spectra look similar, and photochemical production is strongest for both during peak daylight hours. Most AMS factor analysis studies are only able to pick out regional background and locally produced OOA (Jimenez et al., 2009 and references therein). Slowik et al. (2010) identified a biogenic SOA factor by correlation with biogenic VOCs, but noted the mass spectrum was very similar to SV-OOA formed from anthropogenic pollution. As discussed in Sect. 3.2.2, the correlation between OA and CO is indicative of a fossil fuel component of the terrestrial OA, but we also cannot rule out a biogenic component.

### 3.2.4 Interpretation of nitrate measurements

The formation of nitrate aerosol is only possible when there is sufficient ammonium (or other cations) to fully neutralise any sulphate aerosol (Seinfeld and Pandis, 1998). However, our observations in Figure 4 and 5 show the apparent presence of aerosol nitrate under acidic conditions. The measurements are above the detection limit for nitrate (~0.03 µg/sm3 for a 30s averaging time), and are ubiquitous throughout the runs, meaning they are not an artefact of averaging two airmasses of differing composition.

Nitrate appears in the AMS almost entirely in peaks at m/z = 30 and 46 (Allan et al., 2003), for ions $NO^+$ and $NO_2^+$ respectively. When sampling nitrate species, the ratio of these two peaks is determined by the heater temperature and aerosol volatility. For a fixed heater temperature, less volatile nitrate species decompose further during their slower vaporisation, and have a higher 30/46 ratio when measured (Drewnick et al., 2015). The AMS may detect other nitrate species including salts such as $NaNO_3$ and $KNO_3$, as well as organic nitrates, as long as their boiling point is low enough.

Table 4 shows the ratios of m/z 36/40 measured on each flying day during ammonium nitrate calibrations and ambient sampling periods. For 18 July, the 30/46 ratio was the same during the calibration and coastal/inland aerosol. As the ammonium in these periods was sufficient to fully neutralise the aerosol, this gives us a high degree of confidence that the $NO_3$ measured on this flight was ammonium nitrate. However, on all other flights the ambient 30/46 ratio was higher than the calibration values, particularly over the sea. Sea salt-based nitrates may be found in marine air, such as $NaNO_3$ formed by reaction of nitric acid with NaCl. This would be found predominantly in coarse mode aerosol rather than accumulation mode (Harrison and Pio, 1983), but some particles may be detected by the AMS. Alfarra (2004) measured nebulised $NaNO_3$ and recorded a 30/46 ratio of 29.2, which is significantly higher than those measured during COPE. Presumably, the same argument applies with these nitrate salts as with ammonium nitrate: the nitrate would be displaced by sulphate unless the aerosol was fully neutralised. $NaNO_3$ would only be possible if the coarse mode aerosol was not in equilibrium with the accumulation mode. We must therefore investigate the possibility of organic compounds giving signals at the same m/z as the nitrate peaks.

Ovadnevaite et al. (2011) measured high-resolution mass spectra of primary marine OA, which allowed them to identify the specific ions measured at particular m/z values. Around 1.7% of primary marine OA was measured at m/z 30, most of which was $CH_2O$, and 0.2% was at m/z 46. If the measured nitrate concentrations in COPE were attributable to organic species, they would make up 6 – 12% of organic mass in marine air when corrected for relative ionization efficiency. Rollins et al. (2010) measured 30/46 ratios of 0.99 – 5.30 for various organonitrates, which is more in line with our measurements.

To summarise, we are not able to conclusively explain the reported nitrate measurements in acidic conditions. Several past studies have faced similar problems in various different regions e.g. (Allan et al., 2004, 2006, 2014). It is likely that they are partly due to interference from organic species, but the inland 36/40 ratios, where enhanced ammonium concentrations were measured, were closer to those expected from $NaNO_3$. The measurements would need to be performed with a high-resolution mass spectrometer to confirm the organic fraction at the nitrate peaks. There may also be a contribution from sea salt-based nitrate species such as $NaNO_3$, but these are also difficult to quantify with our measurements. On all flights other than 18 July, the total nitrate loadings were a relatively small fraction of the total mass measured by the AMS, as the $SO_4$ and OA loadings were both several times higher.

## 4. Discussion

### 4.1 Interactions between meteorology and aerosol properties

### 4.1.1 Contrasts between marine and polluted airmasses

Of the results outlined in the previous Section, the main contrast in aerosol properties were between the cleaner SWW cases and the pollution event measured on 18 July. In the SWW cases, the aerosol advected to the peninsula from the Atlantic was typical of marine aerosol in terms of its composition and size distribution. The mean aerosol mass loadings measured by the

AMS over the NW coast on these days all agreed within 30%, as did the and the relative fractions of the different chemical species were also similar, with marine sulphate dominating. The CN concentrations over the NW coast (listed in Table 1) were only available on 25 and 29 July, but the integrated SMPS size distributions from Figure 3 (g) also agreed within 30%, and the peak sizes of each of the aerosol modes were also consistent.

In contrast, the easterly winds on 18 July brought anthropogenic pollution to the region, which increased the submicron aerosol mass loadings by an order of magnitude. There were higher loadings of all species, particularly OA, which was the dominant species during the pollution episode, and has a mass spectrum more similar to aged urban emissions compared to the marine OA measured in the other cases. The number concentration of aerosols in the accumulation mode on 18 July was significantly higher than on any of the other days studied, due to the anthropogenic pollution influence, and the CN

concentrations over the NW coast was several times higher than the cleaner cases. The concentration of particles in the inland nucleation mode on 05 July was also higher than on any of the other flights, and this mode was broader than the other flights, which may be due to the stagnant winds allowing photochemical processing to occur within the airmass with minimal mixing of clean marine air. However, most of the pollution is likely to be advected from urban areas further east, rather than emitted locally and allowed to build up, as the southwest peninsula of the UK is not a large source region of

urban pollution. Other than these two flights, where fossil fuel emissions had a larger effect than the other case studies, the size distribution and composition of the accumulation mode aerosol, which is of most relevance to cloud-forming particles, was fairly consistent.

### 4.1.2 Terrestrial and marine influences

Section 3 highlighted some differences observed between aerosol properties in the same airmass measured over the NW

coast or inland. When the cleaner marine airmasses made landfall, the sulphuric acid was partially or fully neutralised to form ammonium sulphate, and in the case of 28 July this persisted to the NW coast measurements which were downwind of the landmass. Agriculture is the largest land use category in the southwest peninsula (Morton et al., 2011), including a mix of both arable and pastoral farming, predominantly cattle grazing on managed grass fields. Previous measurements have found elevated levels of ammonia in the southwest peninsula compared to those found in the Atlantic (Quinn et al., 1996; Sutton et

al., 2001). The terrestrial ammonia emissions are therefore most likely to be of agricultural origin.

By replacing the $H^+$ ions with $NH_4^+$, the neutralisation of the marine sulphuric acid inland should add to the inorganic mass, causing a marginal increase in average particle size. However, the corresponding increase in inorganics was balanced out by similar decrease in $SO_4$ inland, and no consistent differences were observed between the corresponding aerosol size distributions.

One clear feature of the terrestrial influence was the presence of nucleation mode aerosols inland. As shown in Fig. 8a, in all cases the inland CN concentration was significantly higher than that measured over the sea, and higher than typical marine concentrations in the absence of nucleation events (Dall'Osto et al., 2011; Jensen et al., 1996). The aerosol size distributions

also showed concentrations of particles <60 nm were always higher inland. Coastal nucleation is unlikely to explain these measurements, as there was no bias in concentrations towards coastal regions.

It is possible that the formation of these particles is also linked to agricultural emissions. Although particle formation rates from $H_2SO_4$–$NH_3$–$H_2O$ ternary nucleation are lower than those observed in ambient environments, ternary nucleation involving other agricultural emissions (such as amines) may explain our results (Almeida et al., 2013; Pirjola et al., 2000). The high standard deviation/mean ratio of the inland CN concentrations in SWW suggests the process was spatially variable, rather than ubiquitous throughout the boundary layer.

## 4.2. Aerosol influence on CCN and cloud drop concentrations

### 4.2.1 Linking CCN and CDNC

CCN data were only available on 18, 25 and 29 July, and 03 August. Figure 8b shows the average boundary layer CCN concentrations measured on these flights, as well as a comparison to CDNC measured in updrafts near cloud base. Figure 8c shows the in-cloud distribution of vertical wind near cloud base; we use an updraft threshold of 2 m s$^{-1}$, which is around the 75$^{th}$ percentile of cloud base vertical wind on the four flights shown. By choosing only data in updrafts, as well as excluding data from cloud edges, we aim to minimise the influence of entrainment, though it is unlikely to be eliminated. The higher values of CDNC are those least influenced by entrainment, and those with the strongest updrafts when the cloud formed.

The CN and CCN concentrations showed no correlation, and the fraction of CN active as CCN varied from 2% to 50%. This fraction was largest on 18 July, when the average particle size was the largest of all the flights. On all flights, the CCN fraction was lower inland than over the NW coast, particularly in the SWW cases where it was limited to single-figure percentages even at 0.9% SST. As noted in the previous section, the majority of inland particles were smaller than 60 nm and therefore too small to act as CCN. In these cases, the aerosol size distributions were a more important factor than the CN concentrations for determining CCN concentrations.

On 18 July, the peak CDNC was close to the CCN concentration at 0.1% SST. For the SWW flights, the peak CDNC was more comparable to the CCN concentrations measured at 0.9% SST, though the median CDNC was between the two CCN measurements, which is reasonable based on previous estimates of SST in cumulus clouds. For example, Politovich and Cooper (1988) calculated SST up to ~0.3% in cloud passes with a high fraction of air from near cloud base.

Comparing CCN measured inland and over the sea, the differences were generally <10%, with the largest being 30%. There was no clear systematic difference between land and sea. The addition of ammonium inland did not always increase inorganics, as the change was not always larger than the inherent variability in the $SO_4$. On an airmass level, there appears to be an aerosol influence on cloud drop concentrations- the higher aerosol concentrations in the polluted air caused higher

CCN concentrations. This difference was also manifested in the CDNC, which was several times higher in the polluted case on 18 July.

Dall'Osto et al. (2010) reported measurements of aerosol composition and CCN at Mace Head, Ireland during late spring and early summer. Mace Head shares similarities with the southwest peninsula of the UK in that it experiences Atlantic air under the prevailing westerly winds but is also a receptor site for aged urban pollution during easterlies. The CCN concentrations on 18 July were around double those measured by Dall'Osto et al. (2010) in continental pollution, though the aerosol mass loadings here were also significantly higher. In marine airmasses, the CCN concentrations at Mace Head were ~200 cm$^{-3}$ at 0.1% SST and ~400 cm$^{-3}$ at 1.0% SST. These numbers are both in good agreement with our measurements in the SWW cases.

### 4.2.2 Aerosol chemistry, CN and CCN

In marine air, CCN may be produced by sea spray or condensation of secondary aerosols, primarily $SO_4$. These processes can interact, for example primary sea spray subsequently coated with $SO_4$. Particles produced by sea spray may be composed of sea salt, OA, or a combination of the two, with OA more dominant in submicron particles (Cavalli et al., 2004). Sea salt is an effective CCN; OA from sea spray is likely to be insoluble (Gershey, 1983), but may be internally-mixed with sea salt and/or serve as a condensation nucleus for secondary organic or inorganic aerosol formation. During runs over the southwest peninsula, terrestrial emissions may also have influenced CCN concentrations, such as the addition of ammonia or advected aged urban emissions.

As an air parcel cools as it rises, there may be some co-condensation of semi-volatile species such as ammonium nitrate or SV-OOA, which may further enhance CCN concentrations (Topping et al., 2013). As the aerosol inlets on board the BAe-146 do not reliably sample aerosol during cloud penetrations, we are unable to assess how much of an impact this made, but we note that sulphuric acid / ammonium sulphate (the largest aerosol component on most case studies) is not semi-volatile at the temperatures sampled.

As the source functions of sea spray and secondary organic and inorganic marine and terrestrial aerosols all depend on different factors, they would not be expected to be correlated. Therefore, by comparing the measured CCN concentrations to different chemical species, we may gain information on the sources of the CCN.

Figure 9 shows the relationships between CCN concentrations measured aboard the BAe-146 at 0.1% SST and the PCASP number concentration, AMS total mass, non-sea salt inorganics, and OA. Although the validity of the nitrate measurement is unclear on some flights, the nitrate fraction on these days was low. The data in Figure 9 are split between the three SWW case study flights and the polluted case of 18 July. If we did not make this distinction, all the variation and correlation coefficients would be dominated by the difference between the marine and polluted airmasses, rather than variations in concentration within these airmasses.

The correlation between CCN and aerosol number concentration measured by the PCASP was good, and the data fell close to the 1:1 line, meaning the critical diameter was close to the PCASP's lower cutoff of ~120 nm. Good correlations ($R^2 >=$ 0.66) were found in both marine and inland environments comparing CCN with total AMS mass and inorganics. The correlations with OA were lower, but non-zero, and the OA correlation in marine air was better than inland. These results are consistent with the main source of CCN being accumulation mode sulphate-containing particles in both marine and inland air. Liu et al. (1996) found a similar correlation with sulphate at 0.06% SST for cloud-processed aerosol. We also investigated correlations between the x-variables in Figure 9 and the CCN concentrations at 0.9% SST, but the correlations were not as good. This is likely to be because smaller particles activate at 0.9% SST, which are less associated with particle mass and may be smaller than the PCASP's detection range.

While our data suggest that the average CCN were composed primarily of $SO_4$, it does not provide information on the origin of these particles. For example, marine $SO_4$ may condense onto primary sea-spray OA particles, which are largely insoluble when initially formed (O'Dowd and de Leeuw, 2007). Pierce et al. (2007) showed that this type of process can significantly enhance CCN concentrations globally by providing a surface for sulphuric acid to condense on to. Particle emissions from sea spray are strongly linked to wind speed (O'Dowd et al., 1997). This would mean the marine particle concentrations on 29 July would likely be the highest due to the stronger wind speeds, as shown in Fig. 2. While the marine CN concentration on 29 July was higher than on 25 July, it is difficult to draw firm conclusions from such a limited comparison.

Shipping emissions may also serve as condensation nuclei for secondary aerosol (e.g. Langley et al., 2010), but the low BC concentrations in our case studies suggests this was of little importance. Alternatively, new particles may form through nucleation of gas-phase precursors, and open-ocean nucleation events are thought to be most likely to be related to biological activity (O'Dowd et al., 2010). We gain some insight into particle formation by comparing to the size distributions classified as "clean marine" and "open-ocean nucleation" by Dall'Osto et al. (2011), measured at Mace Head. In particular, the value of dN/dlogDp in our lowest size bin (20 nm) is a useful comparison for the smallest particles, which have experienced the least secondary growth. On 05 and 25 July, the values of the lowest size bin were ~100 $cm^{-3}$, which is consistent with the "clean marine" values (~80 – 150 $cm^{-3}$). The concentration on 29 July was higher at 985 $cm^{-3}$, which is closer to that in "open-ocean nucleation" airmasses (~800 – 2500 $cm^{-3}$), but the value on 03 August of 516 $cm^{-3}$ fell between the two categories. It is therefore not clear what contribution primary and secondary aerosol formation had overall to the aerosol number concentration in marine air, but the majority of CCN mass was composed of $SO_4$, which must be secondary.

This leads us to the conclusions that in the SWW cases, the main factor determining CCN concentrations was the size and composition of marine aerosols reaching the peninsula. The majority of accumulation mode aerosol measured by the AMS in these cases was $SO_4$. The measured CCN also exhibited a high degree of correlation with inorganics, which were predominantly $SO_4$ and its cations. If this correlation were due to the mixing of two homogeneous airmasses, the OA would also show a similar degree of correlation with CCN; however, the $R^2$ values for OA were much lower. The data therefore show that whatever process is responsible for the production of sulphate is also largely responsible for determining the

concentration of CCN. Our results are consistent with the prevailing theory that the majority of marine CCN are composed of $SO_4$ formed from DMS oxidation (Charlson et al., 1987).

## 4.3 Sources of INPs

The concentrations of primary ice and the temperatures at which they develop are a large source of uncertainty in the timeline of cloud glaciation. During COPE, the lowest cloud top temperatures were ~-15C (Taylor et al., 2016). Reviews by Hoose and Möhler (2012) and Murray et al. (2012) identified the only species that have been shown to behave as INPs in laboratory experiments at these temperatures as mineral dust; primary biological aerosol particles (PBAP) such as bacteria (Diehl et al., 2006), fungal spores and pollen (Pouleur et al., 1992); and black carbon, although the latter is relatively inefficient. Emissions data for bacteria and fungal spores is limited but global model studies on clouds and precipitation including these species acting as INPs suggest a small but significant zonal influence on precipitation rates (Spracklen and Heald, 2014). Their concentrations at cloud formation levels are generally much lower than mineral dusts, though there is evidence that ice-active biological material internally-mixed with mineral dust and may enhance the ice nucleating potential of the dust (Augustin-Bauditz et al., 2016). Recently, attention has also focused on marine surface organics, which may be aerosolised as sea spray (Burrows et al., 2013).

Mineral dust concentration and size distribution were measured using the BAe-146 filter system, by combining the 'silicates' and 'mixed silicate' categories described by Young et al. (2016). Leon et al. (2015) show a more detailed breakdown of the composition of particles from 03 August. BC and OA concentrations were measured on-line using the SP2 and AMS. There was no instrument capable of isolating PBAP aboard the aircraft, but the ground site hosted a WIBS, which made measurements of UV auto-fluorescent particles. The reader should see Sect. 2 for further details of the instrumentation.

In the absence of INP measurements, parameterisations must be used to calculate INP concentrations. These are empirical relationships derived by comparing measurements of INPs and various chemical species. Figure 10 shows comparisons of several parameterisations, as well as measured first ice concentrations from COPE. The aerosol measurements used for Figure 10 are those from 03 August, as this had the best measurements of ice formation, as presented by Taylor et al. (2016). The measured ice concentrations are from a series of penetrations through a cloud in the early stages of development, when secondary ice processes were thought to have at most a minor influence on ice concentrations (Taylor et al., 2016).

DeMott et al. (2010) based a parameterisation (hereafter D10) for INP concentrations on the total concentration of aerosol larger than 0.5 µm (D >0.5 µm), derived from INP measurements with continuous flow diffusion chambers in a variety of environments around the world. Tobo et al. (2013) formed a similar parameterisation (hereafter T13-T) using data measured in a North American pine forest. We used a D > 0.5 µm concentration of 4.1 $cm^{-3}$ derived from a combination of the PCASP and CDP measurements during the inland aerosol runs. For comparison, the concentration collected on the filters was 3.5 $cm^{-3}$, and calculations performed using this number were not discernibly different when plotted on Figure 10. The INP calculations based on D-10 and T13-T both fall within an order of magnitude of the measured first ice concentrations. The

T13-T trace is closer to the measured ice concentrations than D10, meaning the ice active fraction during COPE was more similar to that in a forest ecosystem than in the average of the environments studied in DeMott et al.(2010).

Niemand et al. (2012) and DeMott et al. (2015) provide parameterisations (hereafter referred to as N12 and D15 respectively) base on mineral dusts. N12 uses the integrated mineral dust size distribution to calculate the number of ice surface active sites using measured size distributions, while D15 again simply uses the number concentration of dust particles larger than 0.5 µm, which in our case was 0.2 cm$^{-3}$ based on the filter measurements. In a previous study, the concentrations of particles collected on the filters agreed well with in-situ probes for particles larger than 0.5 µm, but the concentrations at smaller particle sizes were compared poorly (Young et al., 2016). As D15 uses the dust concentration larger than this size, and N12 is based on the dust surface area distribution, which peaks at supermicron sizes, the filter measurements are expected to be a robust measurement for performing these calculations within the uncertainty of the parameterisations. Both mineral dust parameterisations give INP concentrations several orders of magnitude below the measured first ice concentrations.

As well as T13-T, Tobo et al. (2013) constructed a parameterisation (hereafter referred to as T13-F) based on measurements of fluorescent particles larger than 0.5 µm, which are likely to be PBAP. The measurements upon which this parameterisation was based were made using an ultraviolet aerodynamic particle sizer (UV-APS), which has a different instrument response to the WIBS. Healy et al. (2014) made concurrent measurements using a UV-APS and a WIBS, and found the WIBS channel 3 (which uses the same fluorophore as the UV-APS) was well correlated with the UV-APS fluorescent concentration, but a factor of 2.7 higher due to the instruments' different size ranges.

Between 1100 – 1500 UTC on 03 August, the total concentration of fluorescent particles measured by the WIBS was $100 \pm 40$ L$^{-1}$. The fluorescent concentration measured using channel 3 was $50 \pm 26$ L$^{-1}$, which corresponds to a UV-APS equivalent concentration of $19 \pm 10$ L$^{-1}$. Using this concentration in the T13-F parameterisation generates INP concentrations that are also several orders of magnitude lower than the measured ice concentrations. The difference between the predicted INP concentrations based on T13-T and T13-F are due to the different coarse aerosol composition between the forest environment studied in Tobo et al. (2013) and the marine environment studied here, which is likely to have higher concentrations of sea salt. There may be some difference between the PBAP concentration measured at the ground site to that at the altitude of the aircraft measurements. However, it is unlikely that the PBAP concentration at ~500 m would be three orders of magnitude higher than at ground level, which is what would be required for the INP concentrations calculated using the T13-F parameterisation to agree with the measured ice concentrations.

As an additional estimate of biological INPs, Mohler et al (2008) performed chamber experiments using *pseudomonas syringae* bacteria, and found that the fraction of these bacteria that were active INPs at -10C was 0.5%. Multiplying this fraction by the PBAP concentration measured by the WIBS gives an INP concentration of 0.5 L$^{-1}$ at this temperature. As *pseudomonas syringae* are amongst the most active biological INPs studied thus far (Lorv et al., 2014), this value may be considered an upper estimate for biological INPs at this temperature. Though it is possible to classify WIBS measurements

into different PBAP types, even within simple metaclass discrimination (bacteria, fungal spores and pollen) the uncertainty in the derived concentrations can be large, e.g. up to a factor of 5 for bacteria, depending on the assumptions used in the analysis approach, as discussed by Crawford et al. (2015).

To date there are no INP parameterisations for soot in the relevant temperature range, and laboratory studies show mixed results, though it is likely that soot particles are less efficient INPs than mineral dust (Hoose and Möhler, 2012). The total mineral dust mass concentration measured was 4 $\mu$g m$^{-3}$, whereas the total BC concentration was 30 ng m$^{-3}$, so the INP concentrations from BC were likely to have been several orders of magnitude lower than mineral dust.

Wilson et al. (2015) recently published a mass-based parameterisation for marine surface organics measured in the European Arctic. To our knowledge, this is the only parameterisation published for this type of INP. Our results in Figure 10 show that aerosolised marine surface organics are likely to have produced INPs in similar concentrations to mineral dust and PBAP.

In summary, both parameterisations based on total aerosol larger than 0.5 $\mu$m produced INP concentrations of a similar order of magnitude to the measured first ice concentrations. However, all the composition-specific parameterisations produced INP concentrations several orders of magnitude lower. This therefore leaves us with two possible conclusions: either there was a source of INPs that was not characterised by our measurements (i.e. a species other than PBAP, mineral dust and sea spray OA that were significantly active INPs at -5 > T > -15°C), and/or one or more of the composition-specific parameterisations underestimate INP concentrations when used in the southwest peninsula of the UK. The measured ice concentrations lie in the temperature range at the high end of most of the parameterisations tested. For example, the data used to generate the fit parameters for T13-F included just seven data points at -10 °C, and the majority of points were from temperatures ≤-20 °C. Mason et al. (2015) also noted the poor performance of INP parameterisations at a coastal site, and noted the good correlation of PBAP with INPs between -15 and -25 °C in air from the Pacific Ocean. It is possible the sources of INPs are similar between the two studies, but without further characterisation of INPs active at temperatures higher than -15°C, we are limited to such speculation.

## 5. Summary and conclusions

We have presented measurements of boundary layer aerosol concentration, size distribution and composition from a series of research flights performed over the southwest peninsula of the UK during the COPE campaign of summer 2013. Several case studies showed marine air from the Atlantic Ocean advecting to the peninsula from the south/southwest. The CCN and submicron aerosol in these cases were predominantly sulphate-based, and typical for Atlantic marine air. During Easterly winds, polluted air from the UK brought urban pollution to the region; the submicron aerosol was composed of ammonium nitrate and organics, and CCN concentrations were several times higher than the marine cases. In one marine case, INP concentrations were estimated using by inputting field measurements into several size- and composition-specific

parameterisations. Bulk parameterisations predicted INP concentrations within an order of magnitude to measured first ice concentrations, but attempts to identify the composition of the INPs were largely inconclusive.

The periods of southwesterly winds, which brought marine air from the Atlantic to the peninsula, are of particular interest as they are the most conducive to convective cloud formation in the region. The CCN concentrations in these conditions were

~150 – 280 cm$^{-3}$ at 0.1% SST, and 400 – 500 cm$^{-3}$ at 0.9% SST, which are in good agreement with previous measurements in Atlantic air performed at Mace Head (Dall'Osto et al., 2010). The marine aerosol size distributions were also similar to those measured at Mace Head during "clean marine" conditions (Dall'Osto et al., 2011), though some flights had enhanced concentrations of nucleation mode particles, the source of which is not clear from our measurements.

The majority of accumulation mode mass was composed of $SO_4$ on these days, which is likely to be from biogenic DMS

oxidation (Charlson et al., 1987). The high degree of correlation between CCN and SO suggests the CCN were predominantly composed of $SO_4$, though the origin of these particles (i.e. coated primary or nucleated secondary particles) is again unclear. Accumulation mode OA had a similar mass spectrum to previous measurements of sea spray OA (Ovadnevaite et al., 2011), but a low correlation with CCN, meaning sea spray OA that was not internally-mixed with $SO_4$ was not a large source of CCN.

By comparing aerosol over the sea to that measured inland, we infer the influence terrestrial emissions had on the marine aerosol. The marine $SO_4$ was acidic over the sea, but was neutralised by terrestrial ammonia, which is likely to have been emitted by agricultural activities that are widespread throughout the peninsula. The increase in inorganics by this process was almost equally balanced out by a decrease in $SO_4$. Consequently, the differences between oceanic and inland CCN concentrations were less than the day-to-day variability in different marine cases. Additionally, large enhancements of

nucleation mode particles were measured inland, which may also be related to agricultural emissions interacting with marine $H_2SO_4$, but these particles were too small to affect CCN concentrations.

The largest difference in aerosol mass loadings and CCN were between the marine flights and a case study where polluted urban emissions advected to the site from the east. The CCN concentrations were ~6 times higher in this airmass, and the inorganics were composed of equal amounts of ammonium nitrate and sulphate. The difference in CCN also corresponded to

differences in the cloud base CDNC. In the SWW cases the peak CDNCs were ~450 cm$^{-3}$, which is in agreement with the CCN measured at 0.9% SST, though most values of CDNC at cloud base were closer to 300 cm$^{-3}$, which suggests the SST at cloud base was typically between 0.1% and 0.9%. In the polluted case, the peak CDNC was ~1300 cm$^{-3}$, which was in agreement with CCN measured at 0.1% SST in this case.

Possible sources of INPs were assessed by comparing various different parameterisations, using our measured aerosol

concentrations as input. We estimated INP concentrations using measurements of total aerosol larger than 0.5 µm (DeMott et al., 2010; Tobo et al., 2013), mineral dust (DeMott et al., 2015; Niemand et al., 2012), fluorescent aerosol (Möhler et al., 2008; Tobo et al., 2013) and sea spray OA (Wilson et al., 2015). The parameterisations based on total aerosol produced INP concentrations that agreed with measured first ice concentrations within an order of magnitude.

In contrast, the predicted INP concentrations derived using all of the composition-specific parameterisations were several orders of magnitude lower. Either there was a source of INPs that was not characterised by our measurements, and/or one or more of the composition-specific parameterisations tested greatly underestimated INP concentrations when used in the southwest peninsula of the UK. The lowest cloud-top temperatures during COPE were -15 °C. Better characterisation of INPs at these relatively warm temperatures would be required to gain any further insight into the sources of INPs during COPE. It is, however, very unlikely that the ice concentrations of up to several hundred per litre reported by Taylor et al. (2016) could be achieved without invoking the action of secondary ice processes.

The results presented in this paper serve as information for modellers simulating cloud-aerosol interactions for the COPE project, providing details on the aerosol size, composition, CCN and INPs in the different case studies. Our data also provide characterisation of Atlantic aerosol, and are likely to be particularly applicable to regions where marine air makes landfall in the absence of strong fossil fuel emissions.

**Acknowledgements**

The authors wish to thank all those involved in COPE. The BAe-146-301 Atmospheric Research Aircraft was flown by Directflight Ltd and managed by the Facility for Airborne Atmospheric Measurements (FAAM), which is a joint entity of the Natural Environment Research Council (NERC) and the Met Office. Processed data are available from the British Atmospheric Data Centre. Raw data are archived at the University of Manchester and available on request. COPE was supported by NERC under grant numbers NE/J022594/1 and NE/J023507/1, and the Met Office funded the operation of the BAe-146 FAAM aircraft. We also thank Jurgita Ovadnevaite and colleagues from the National University of Ireland, Galway for providing the sea spray mass spectrum. Finally, we thank the editor and two anonymous referees for helping improve the scientific quality of the manuscript.

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

Table 1- Mean aerosol properties measured in NW Coast and inland on each of the case study flights. Where quoted, uncertainty values are standard deviations. All measurements are corrected to standard temperature (273.15 K) and pressure (1013.25 hPa). On 18 July, the CCN column 2 was set to 0.6% SST and 0.9% for all other flying days. Standard deviations are listed for the CCN and CN data, as they are representative of real variations that are much larger than the instrument uncertainty.

| Date | Flight number | Run altitude (m) | Run length (min) | $SO_4$ (µg $m^{-3}$) | Org (µg $m^{-3}$) | MSA (µg $m^{-3}$) | $NO_3$ (µg $m^{-3}$) | $NH_4$ (µg $m^{-3}$) | BC (ng $m^{-3}$) | CCN 0.1% SST ($cm^{-3}$) | CCN Column 2 ($cm^{-3}$) | CN ($cm^{-3}$) | $N_{GA}$ ($cm^{-1}$) | $N_U$ (L⁻ |
|---|---|---|---|---|---|---|---|---|---|---|---|---|---|---|
| **NW Coast** | | | | | | | | | | | | | | |
| 05 July 2013 | B786 | 330±140 | 12 | 2.19 | 1.27 | 0.10 | 0.11 | 0.11 | 12 | - | - | - | 1.3 | 3 |
| 18 July 2013 | B788 | 612±14 | 9 | 5.0 | 11.88 | 0.32 | 5.15 | 3.55 | 530 | 1540±150 | 2860±110 | 5030±220 | 0.71 | 3 |
| 25 July 2013 | B789 | 388±16 | 9 | 2.34 | 0.55 | 0.13 | 0.06 | 0.16 | 10 | 230±21 | 500±100 | 990±150 | 0.83 | 3 |
| 28 July 2013 | B790 | 382±13 | 8 | 1.31 | 0.96 | 0.04 | 0.17 | 0.48 | 26 | - | - | - | 1.4 | 4 |
| 29 July 2013 | B791 | 396±22 | 8 | 1.48 | 0.52 | 0.05 | 0.07 | 0.28 | 18 | 168±25 | 440±140 | 1400±400 | 1.6 | 12 |
| 03 Aug 2013 | B792 | 405±23 | 8 | 1.73 | 0.32 | 0.09 | 0.07 | 0.18 | 7 | 146±9 | 450±120 | - | 1.2 | 6 |
| **Inland** | | | | | | | | | | | | | | |
| 05 July 2013 | B786 | 540±80 | 64 | 1.56 | 2.62 | 0.18 | 0.71 | 0.8 | 97 | - | - | - | 1.60 | 4 |
| 18 July 2013 | B788 | 615±11 | 20 | 4.0 | 10.13 | 0.28 | 4.55 | 3.28 | 470 | 1230±120 | 2740±160 | 7200±800 | 0.75 | 3 |
| 25 July 2013 | B789 | 570±70 | 39 | 1.1 | 0.67 | 0.10 | 0.17 | 0.67 | 32 | 278±27 | 500±50 | 8000±6000 | 0.7 | 3 |
| 28 July 2013 | B790 | 520±30 | 18 | 1.1 | 0.97 | 0.04 | 0.23 | 0.5 | 30 | - | - | - | 0.9 | 4 |
| 29 July 2013 | B791 | 530±40 | 20 | 1.4 | 0.54 | 0.05 | 0.1 | 0.39 | 31 | 164±49 | 410±120 | 9000±5000 | 1.1 | 4 |
| 03 Aug 2013 | B792 | 550±40 | 20 | 1.2 | 0.42 | 0.08 | 0.09 | 0.49 | 30 | 212±33 | 440±70 | 7000±2000 | 0.7 | 3 |

Table 2 - Mean diameter of distinct aerosol modes measured over the NW coast and inland on the different cast study flights.

| Date | Mean diameter (μm) | | |
|------|----------------|--------------------|----------------|
| | Aitken mode | Accumulation mode | Coarse mode |
| **NW Coast** | | | |
| 05 July | 0.048 | 0.188 | 3.92 |
| 18 July | N/A[a] | 0.125 | 3.25 |
| 25 July | 0.060 | 0.184 | 4.18 |
| 28 July | 0.049 | 0.153 | 3.04 |
| 29 July | 0.041 | 0.155 | 4.20 |
| 03 August | 0.043 | 0.158 | 4.00 |
| **Inland** | | | |
| 05 July | N/A[a] | N/A[a] | 3.71 |
| 18 July | N/A[a] | 0.114 | 3.13 |
| 25 July | 0.055 | 0.169 | 3.76 |
| 28 July | 0.051 | 0.145 | 3.56 |
| 29 July | N/A[a] | 0.151 | 3.61 |
| 03 August | 0.057 | 0.155 | 3.87 |

[a]Mode not distinct from surrounding distribution

Table 3 - Uncentred R (also known as normalised dot product) between marine OA measured on the SWW case studies and several literature spectra. m/z 28 was not used for the correlations due to gas-phase interference.

| | Uncentred R | | | | | |
|---|---|---|---|---|---|---|
| Date | Sea spray (Ovadnevaite et al., 2011) | Marine biogenic SOA (Chang et al., 2011) | Organic factor (Chang et al., 2011) | HOA (Morgan et al., 2010) | OOA-1 (Morgan et al., 2010) | OOA-2 (Morgan et al., 2010) |
| 05 July | 0.99 | 0.84 | 0.99 | 0.47 | 0.81 | 0.83 |
| 25 July | 0.97 | 0.89 | 0.97 | 0.51 | 0.78 | 0.83 |
| 29 July | 0.99 | 0.79 | 0.98 | 0.38 | 0.83 | 0.79 |
| 03 August | 0.95 | 0.85 | 0.96 | 0.46 | 0.79 | 0.79 |

Table 4 – Comparison of the m/z 30/46 ratios measured in the AMS aboard the BAe-146.

| Date | (m/z 30) / (m/z 46) ratio | | |
|---|---|---|---|
| | $(NH_4)NO_3$ calibration | NW Coast | Inland |
| 05 July | 1.77 | 6.33 | 2.78 |
| 18 July | 1.65 | 1.67 | 1.66 |
| 25 July | 1.48 | 6.61 | 3.05 |
| 28 July | 1.51 | 4.40 | 3.14 |
| 29 July | 1.37 | 7.84 | 6.15 |
| 03 August | 1.33 | 8.68 | 4.01 |

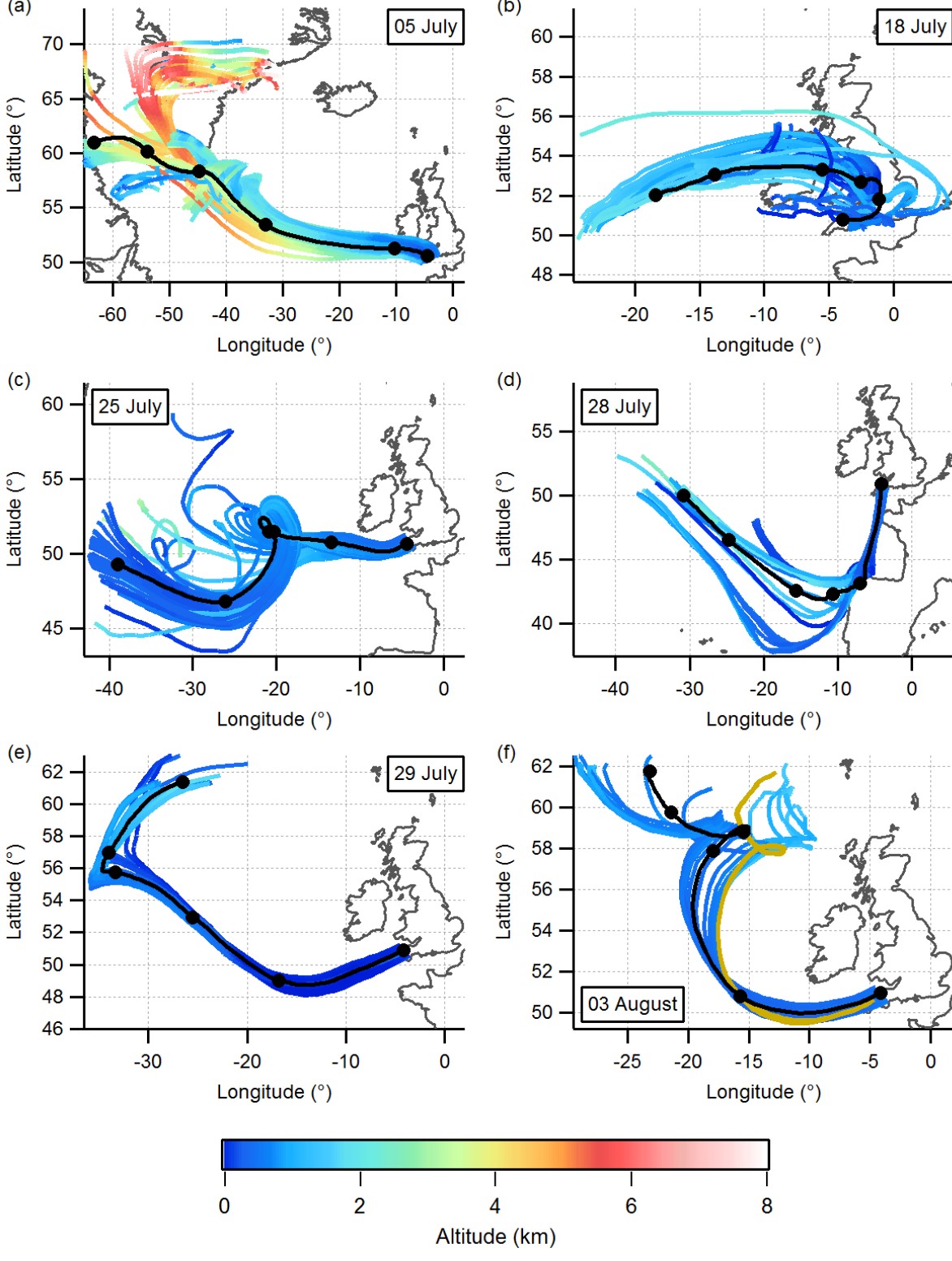

Figure 1 - 5-day HYSPLIT back trajectories initiated during the boundary layer aerosol runs. Panels (a) – (f) show the back trajectories on different flying days. Individual trajectories are coloured by altitude. The black lines show the mean trajectories, with markers every 24 hours before sampling. The two gold lines in panel (f) (which lie approximately on top of each other on the plot) are the two trajectories released from the ground site at 1100UTC and 1200UTC. The altitude of these two trajectories remained below 200 m in the first 48 hours, and below 1000 m in the 120 hours of the runs.

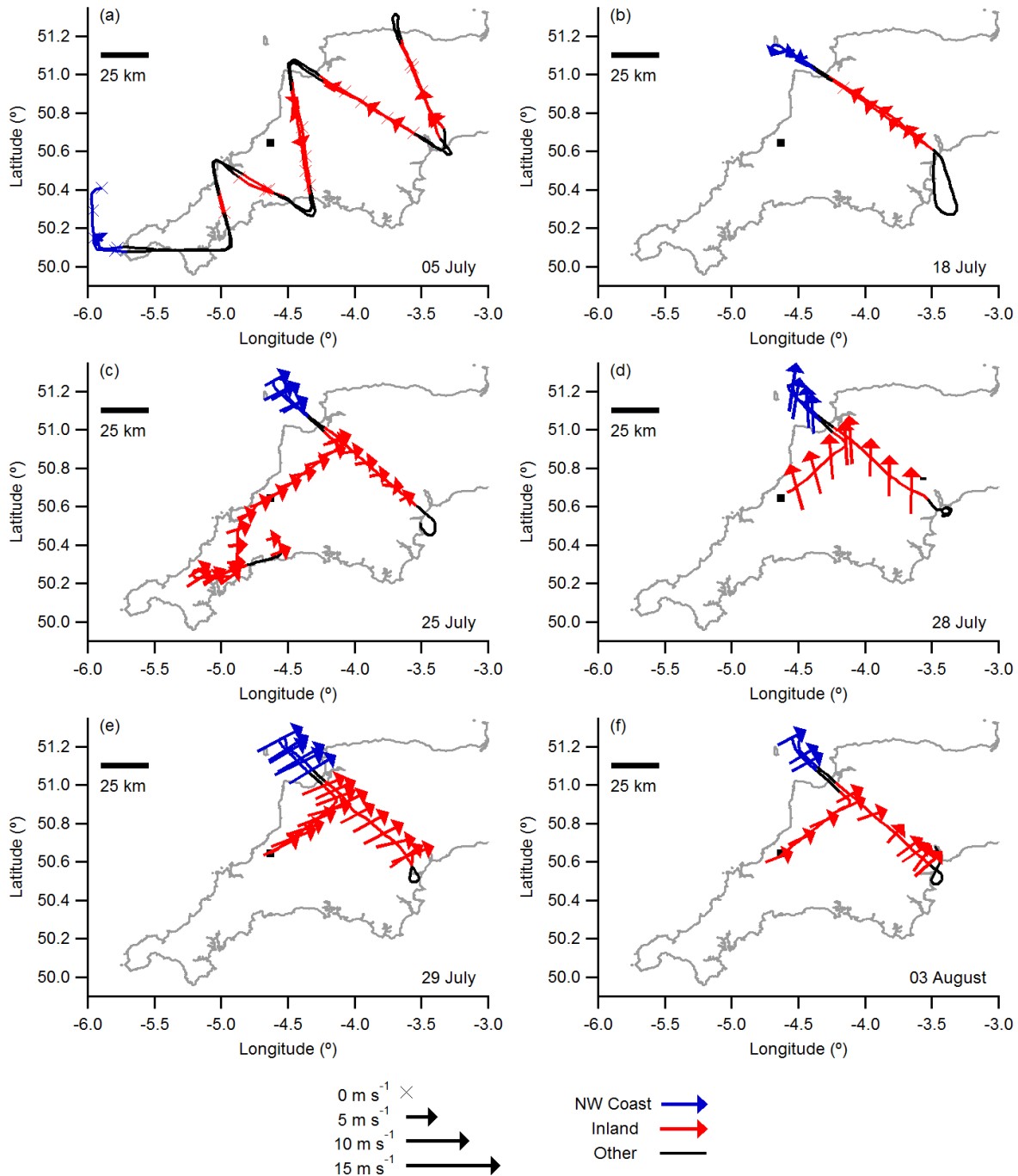

Figure 2 - Spatial location of the below-cloud aerosol runs on each case study flight. Panels (a) – (f) show the six case study flights. The arrow markers show the measured wind speed and direction. The black square shows the location of the Davidstow ground site.

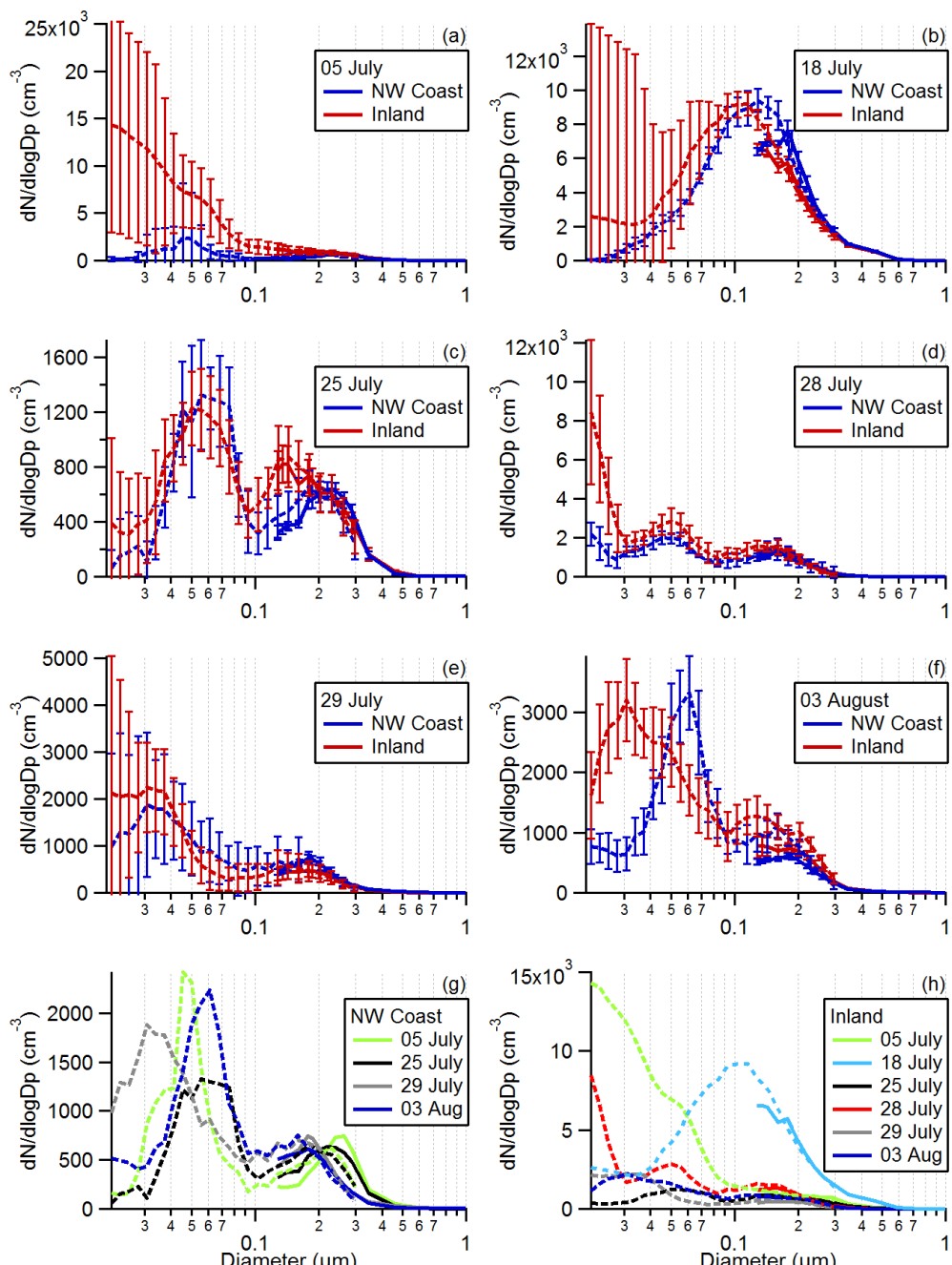

Figure 3 – Submicron aerosol size distributions measured by the SMPS (dashed lines) and PCASP (solid lines) probes aboard the BAe-146. Panels (a) – (f) show the inland and NW coast distributions for each of the case study flights, while panels (g) and (h) show comparisons of all the NW coast and inland distributions respectively. The error bars in panels (a) – (f) show the standard deviations over each measurement period.

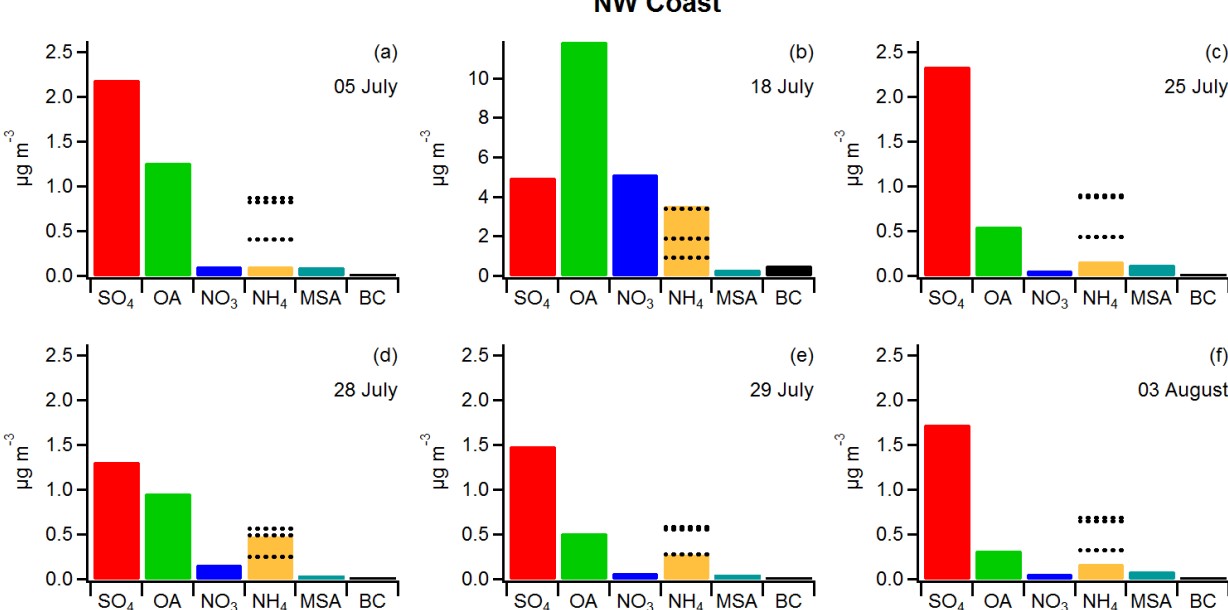

Figure 4 - Aerosol composition measured over the NW coast during the six case study flights. The definitions of each chemical species are given in the text. The lower and middle dashed lines show the concentration of $NH_4$ required to neutralise the $SO_4$ to ammonium bisulphate and sulphate respectively. The top dashed line shows the concentration of $NH_4$ required to fully neutralise all measured inorganic components. The reader should consult the text for discussion of the $NO_3$ measurements.

**Inland**

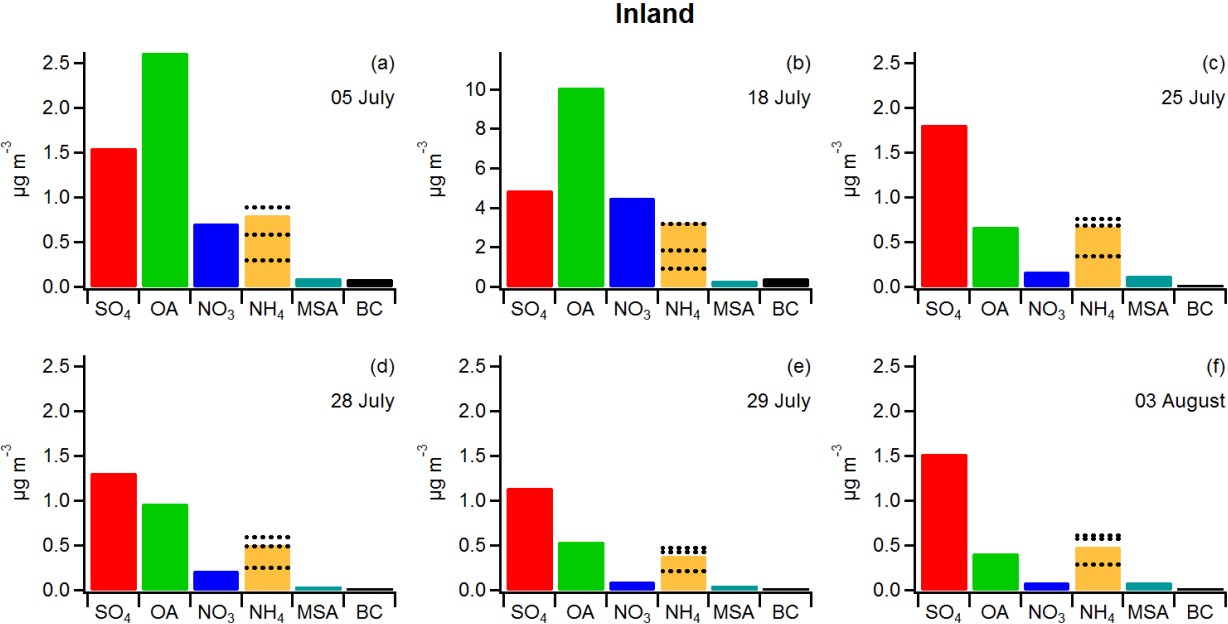

Figure 5 - As in Fig. 4 but for measurements taken inland over the southwest peninsula of the UK.

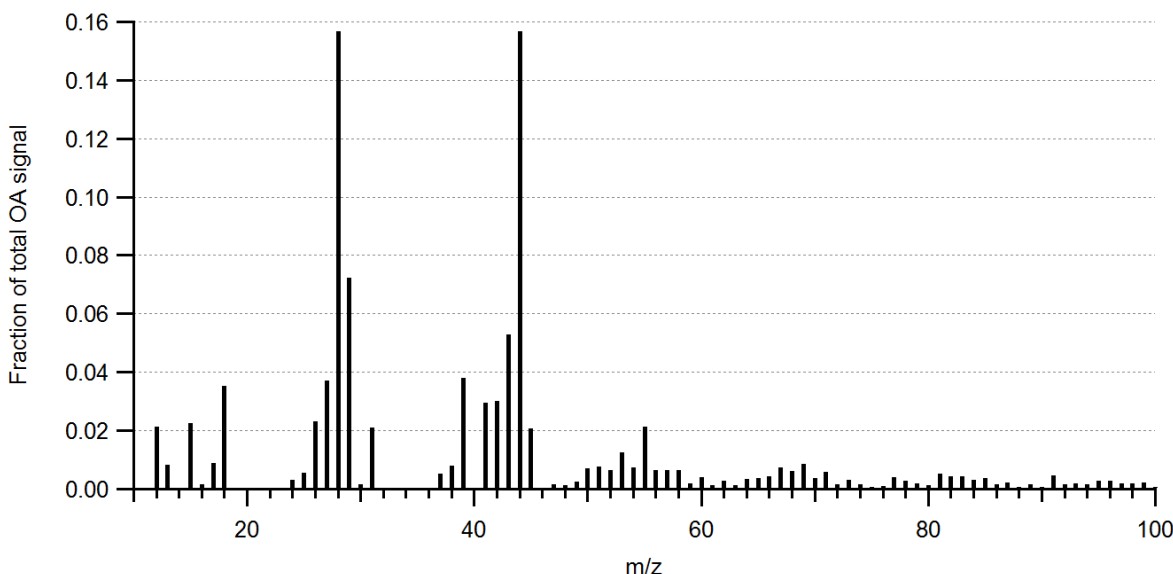

Figure 6 – Normalised mass spectrum of marine OA measured over the NW coast on 05 July.

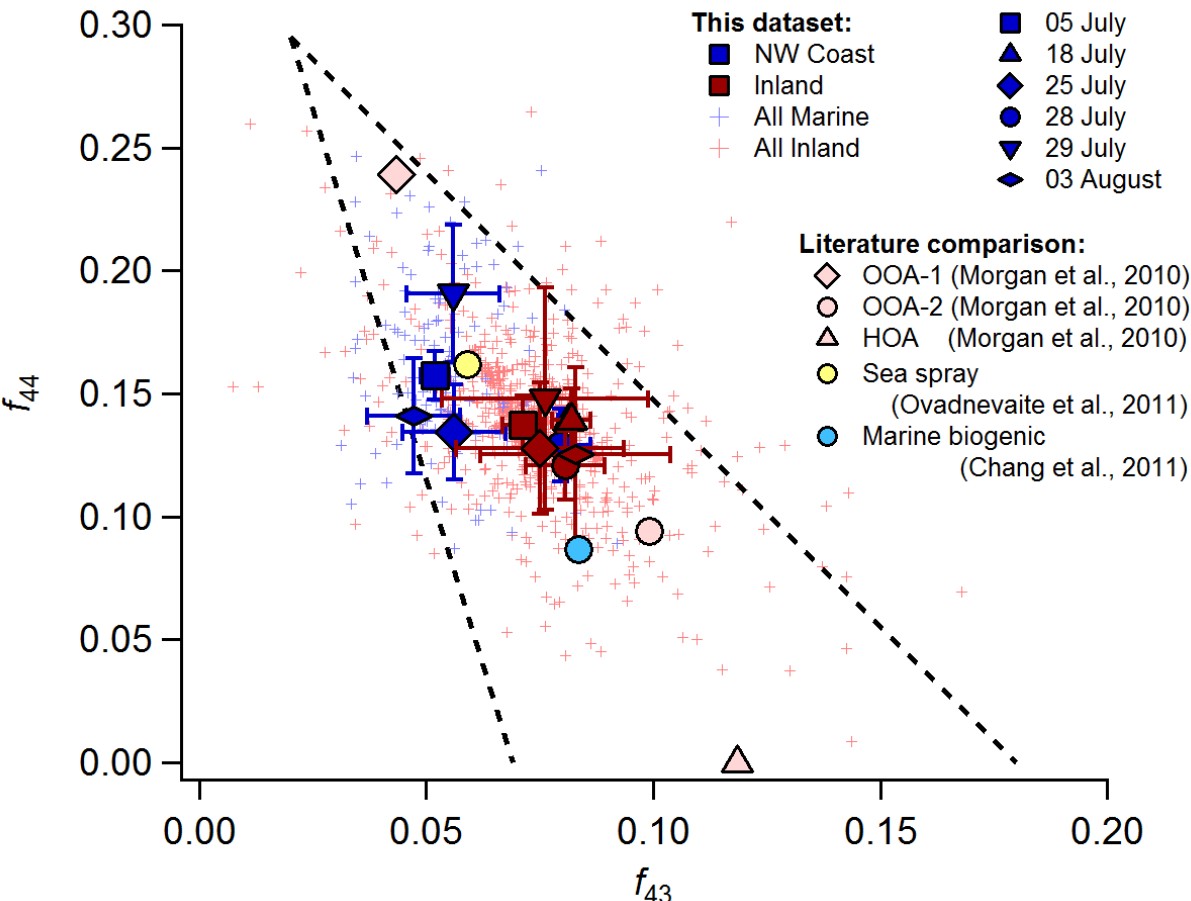

Figure 7 - Relationship between the fraction of OA signal at (m/z) = 44 and 43, in coastal and inland areas. The diagonal dotted lines define the region where ambient OOA measurements tend to fall (Ng et al., 2010). The data points labelled "All marine" are the NW coast data from 05, 25 and 29 July and 03 August, whereas the points labelled "All inland" are from all six flights. The literature values are the same as those in Table 3.

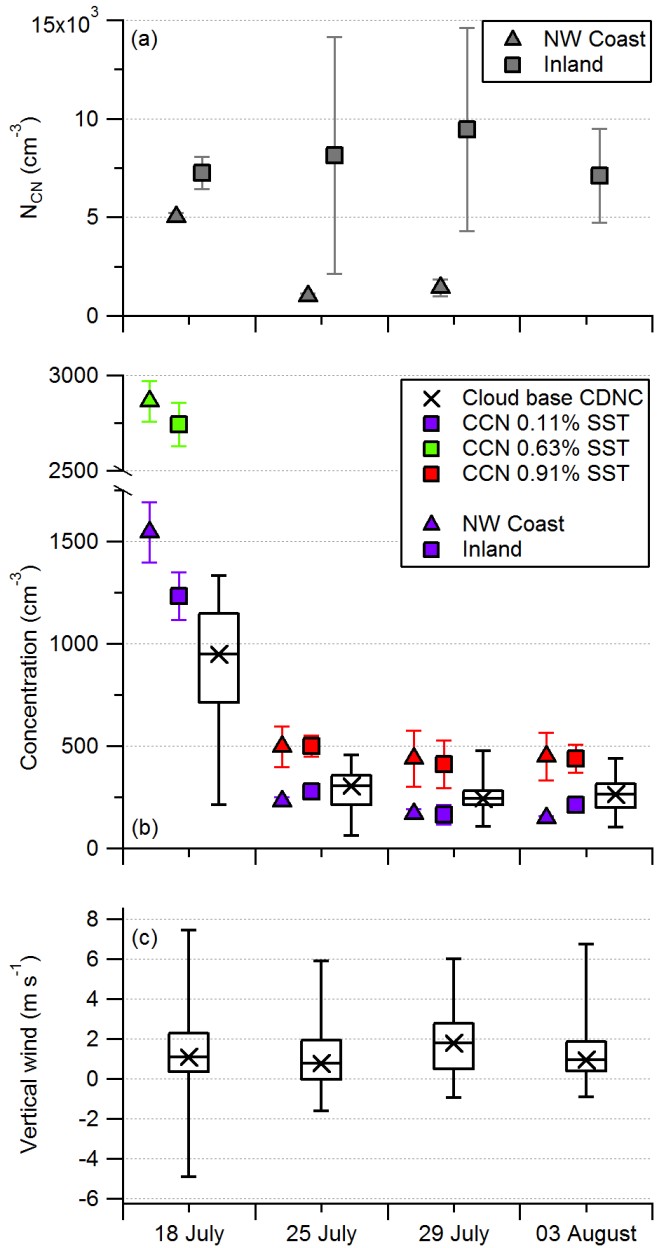

Figure 8 – Comparison of particle concentrations measured in marine air (triangles) and inland (squares). Panel (a) shows the CN concentrations, while panel (b) compares the CCN concentrations to the CDNC measured in updrafts >2 m s$^{-1}$ within 500 m of cloud base. The boxes and whiskers show the 25/75$^{th}$ and the maximum/minimum. Panel (c) shows the distribution of vertical wind in cloud penetrations (liquid water content > 0.01 g m$^{-3}$) within 500 m of cloud base.

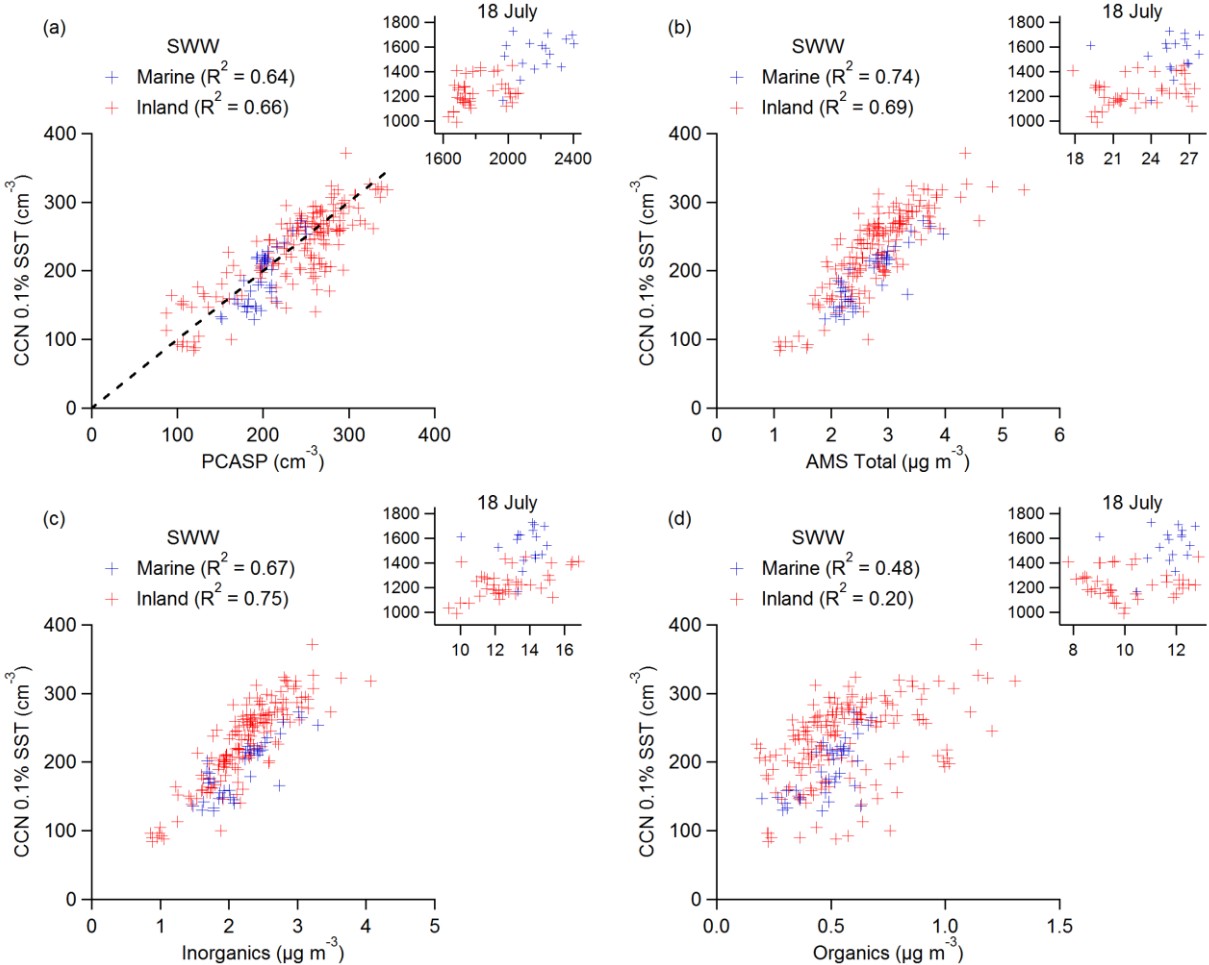

Figure 9 - Scatter plots of measured CCN concentration at 0.1% SST with concentrations of aerosol species. The data shown are from 18, 25 and 29 July, and 03 August, as CCN data were only available on these days. The blue data points were measured over the NW coast, and the red points were measured inland, as defined in Figure 2. Data from the SWW case and anthropogenic pollution (18 July) cases are separated due to the different meteorological conditions and greatly differing CCN concentrations.

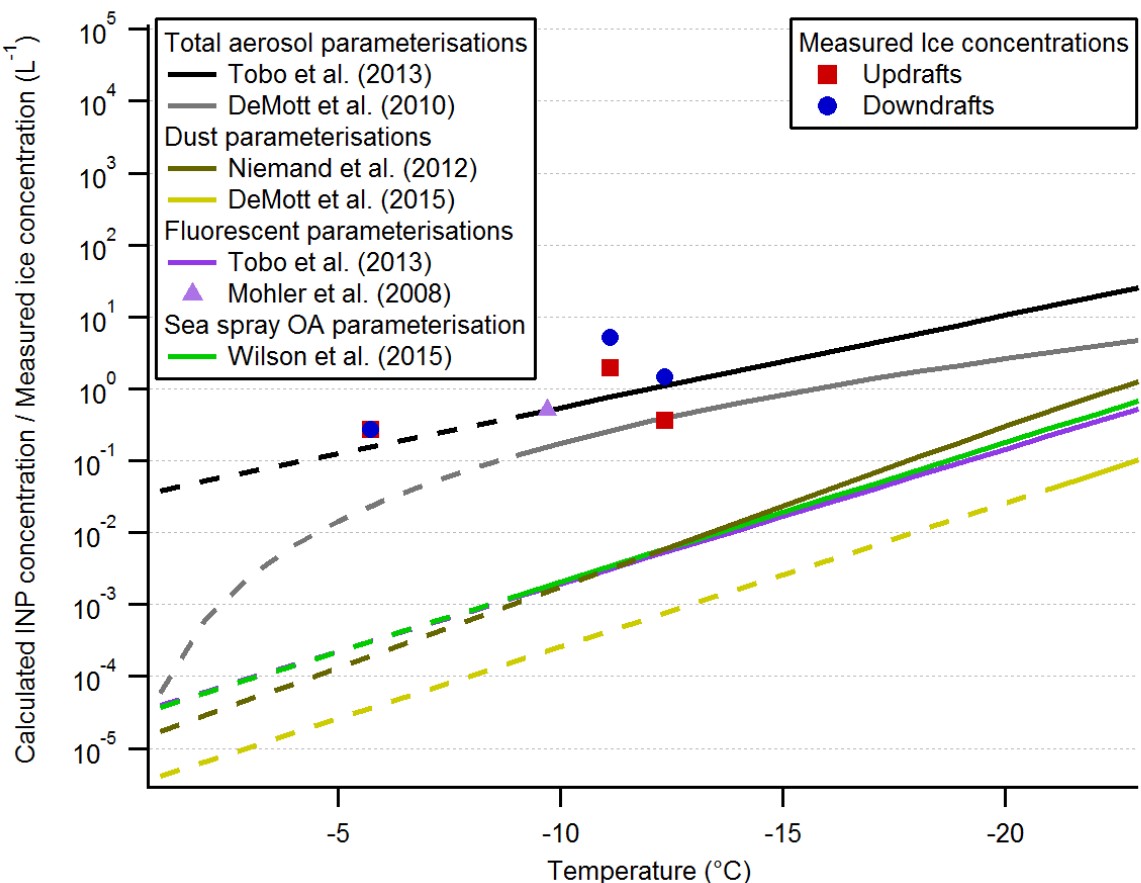

Figure 10 - Comparison of INP calculations using different parameterisations. The solid lines show the temperature ranges in which the INP parameterisations are deemed valid, while the dashed lines are extrapolations. The inland aerosol measurements were used as input for the parameterisations, other for Wilson et al. (2015) where the marine OA concentration was used. The blue and red markers show measured first ice concentrations from Taylor et al. (2015) at different cloud top temperatures.