# Peer review of "Aerosol measurements during COPE: composition, size and sources of CCN and INPs at the interface between marine and terrestrial influences"

_Atmospheric Chemistry and Physics, 2016_

## Referee Comment (RC1) · Anonymous Referee #1 · 16 Mar 2016

The study investigates cloud condensation nuclei (CCN) and ice nucleating particle (INP) characteristics in the marine-terrestrial region over the southwest peninsula of the UK in 2013. A comprehensive data set of aerosol chemical and microphysical parameters was acquired during several flights. One of the main results includes specific CCN number concentrations at 0.1 and 0.9 % supersaturation during clean and polluted conditions. Another result is that significant knowledge gaps still exist in predicting INP concentrations as the comparison of INP concentrations with existing parameterizations shows. The paper is written clearly, and the research and data analyses have been carried out very carefully with great attention to details. This work is a worthwhile contribution to the body of literature on CCN and INP, especially with respect to

the marine-terrestrial interface and the differences between natural and anthropogenic CCN and INP contributions. I recommend publication with minor revisions.

General comments:

The authors introduce the greater topic of convective cloud formation and linkages to flash floods on the southern UK. However, the actual results of the COPE study are not interpreted in relation to this theme. To further improve this work, I recommend adding a more direct description of how COPE relates to convective cloud formation in the introduction, and a respective paragraph in the discussions section.

As far as I am aware, ice nuclei are referred to ice nucleating particles (INP) in the current literature. I recommend changing this throughout the manuscript.

Specific comments:

p. 1, l. 16: specify if the concentration are given for STP or ambient conditions.

l. 20: What was the supersaturation at cloud base?

l. 22: It is not clear what the authors mean by "Marine organic aerosol (OA) had a similar mass spectrum to sea spray". What type of spectra are you comparing: only OA spectra in marine air masses to only sea spray spectra? I would expect that the OA spectrum does not show fragments of sodium and chloride (and other salts) while the sea spray spectrum is dominated by salts. Some clarification is needed.

p. 2, l. 2: The region is not particularly vulnerable because of convective cloud formation and flash floods. The region is particularly vulnerable because there is human activity where flash floods occur. In other words, if nobody lived there, there would be no vulnerability. Rephrase.

p. 3, l. 14f: Include a reference or explanation for the quantification of the uncertainty.

l. 19: a reference is missing.

Section 2.1 Aircraft Measurements: Information about the inlet system is missing. Did you use a pressure controlled inlet? If yes, for which instruments, if not how did you treat/correct the data?

p. 4, l. 6: which densities did you use to calculate the volume?

p. 5, l. 4: As far as I am aware, the recommendation is to use the latest reference for HYSPLIT as noted on their webpage. Further regarding the back trajectory analysis, did you run individual trajectories only or ensembles for each release time? If you did not run ensembles, how can you guarantee that the single trajectory is representative and accurate enough? Some more information on how you performed the analysis and why is needed.

l. 19: I would not state "any long-range transported" aerosol was washed out, but rather "most". Air masses that have experienced precipitation are not completely free from aerosol.

p. 6, l. 24: Information is missing on how you determined nss-chloride. We know from the literature you cite and further references that the AMS does measure a fraction of sea salt. So the signal at the chloride fragment m/z must be partially originating from sea salt.

p. 10, l. 5f: Was there no coarse mode from sea salt that was measured? More complete information on the size distribution is needed here.

l. 7: From what I see in the figure, I cannot agree with the statement that all distributions were open at the lower end, e.g. in panel f, the red curve is not open (also in others). This needs some more detailed discussion. In case I misinterpret the meaning of "open" a clearer description is needed.

p. 13, l. 4-9: Is there any particular reason why you did not apply kappa-Köhler theory? It seems only logical to derive a kappa value from the AMS measurements to compare it with previously determined kappa values.

l. 29: The first factor determining CCN concentrations is the CN number concentration rather than the size distribution.

Fig. 1: I suggest to use the color code for altitude information and to insert numbers to indicate the age of the air mass.

Technical comments:

p. 1, l. 28: The sentence doesn't make sense grammatically.

p. 3, l. 8:. "." Is missing after "al", also in some other occasions

p. 10, l. 3: There is a grammatical error in the sentence.

Table 1: Include information on the year.

Fig. 7: include information on what the solid and dashed lines represent in the captions.

Fig. 8: A legend explaining what the rectangles and triangles are would help the reader to understand the message more easily and quickly.

Fig. 9: The small graphs in each panel are not explained.

---

## Referee Comment (RC2) · Anonymous Referee #2 · 3 May 2016

Taylor and coauthors provide a noteworthy and comprehensive set of aircraft observations of aerosols, specifically those which serve as cloud condensation nuclei (CCN), over variable source regions during the COPE field campaign. Additionally, they evaluate air mass sources and predict ice nucleating particle (INP) concentrations based on a set of different ice nucleation parameterization models. Although this work represents a detailed account of observations necessary to improve climate model simulations, there are a few issues that need to be resolved prior to publication in ACP.

**General comments:**

In attempts to harmonize ice nucleation terminology, Gabor Vali and colleagues published a technical note to define terms used throughout the community. Please use ice nucleating particles (INPs) instead of IN, to align with Vali et al. (2015).

Vali, G., et al. Technical Note: A proposal for ice nucleation terminology. Atmospheric Chemistry and Physics 15.18 (2015): 10263-10270.

Information regarding the instrumentation used is missing. First, what aircraft inlet was used? I am assuming from a brief statement later on in the text regarding not being able to sample in-cloud that an isokinetic inlet was used, but please provide the details in the methods section. Also, what instruments were used to measure CO, vertical velocity, and ice concentration? What are the units for these? Observations of CO and vertical velocity are presented in the manuscript, but information on the instrumentation is not provided. This would be alleviated by providing a few quick sentences in the methods, perhaps in a supporting measurements paragraph.

The altitude of the measurements and flight duration are vital pieces of information that should be provided. The authors do state the flights occurred below ~500 m, but measurements nearest to the surface could vary significantly from 500 m, depending on stratification, emissions, winds, etc. Even if provided by Leon et al. (2015), that information should be provided again here. This would also provide vertical context to the air mass trajectories as trajectories ending at 100 versus 500 m could lead to disparate sources. The authors could address this by providing an image of the vertical sampling statistics, or even a range if sampling was only conducted at 500 m (this is not currently clear).

Along these lines, on page 5, lines 19-20, the authors suggest that aerosols from long-range transport were likely removed via precipitation, but that depends on how high the trajectories were back in time. For instance, the height of a trajectory endpoint 3-4 days back could be thousands of meters. This statement would be more valid if vertical profiles of the trajectories were shown, perhaps as separate panel(s) in Figure 1. Also, what about trajectories at the ground site, since the authors present data from this location? A connection between the ground and location of aircraft observations would result from also running trajectories where the WIBS measurements were acquired.

The $N_{GA}$ and $N_{UGA}$ data and discussion are not pertinent to the main themes of the manuscript. Further, the values are quite low; are these values of any significance compared to total CCN

concentrations? I suggest eliminating this information as it is only discussed in one small paragraph and does not add any significance to the main conclusions. What the authors could instead do is provide mean size for each day, perhaps for each size distribution mode, to provide some sizing context to the CCN.

Are there enough sampling statistics to look at vertical profiles of the aerosol measurements? That can provide quite a bit of insight into the sources and transport of aerosols. At the very least, this can be done with the SMPS, PCASP, and CDP. Also, RH, mixing ratio, temperature profiles would be helpful to show.

Why is SEM-EDX introduced in the methods, but no results are provided? The single particle chemistry would serve as quite useful information for characterizing the aerosol.

Given the temperature range presented in the last section, I would expect the ice particle number concentration to equate to several orders of magnitude higher than INP concentrations (i.e., Hallett-Mossop). The authors should revise this section to include secondary ice formation as a plausible reason and tone down the element of surprise.

**Specific comments:**

The abstract could use a sentence or two regarding the broader impacts and motivation for the work. By adding some "big picture" material, the significance of this work is evident right off the bat.

Introduction: Similar to the abstract, end the first paragraph with a direct statement to segue into the next paragraph, i.e., something along the lines of, "The potential of flooding from persistent convective clouds along the peninsula demonstrate the importance to understand cloud formation in this region."

As I and potentially other readers are not familiar with this region, it would be helpful to point out Figures 1 and 2 in the first paragraph for geographical context.

Page 1, line 27: Define that the parameterizations are for prediction of ice nucleating particle concentrations. Parameterizations is somewhat vague.

Page 2, line 13: Can also inhibit cold precipitation by reducing riming efficiency of descending ice particles in a mixed-phase cloud system. Also, replace "lower" with "subzero".

Page 2, line 14: "..such as riming and the Hallett-Mossop rime-splintering processes." Also, the aerosols themselves do not "initiate" secondary ice formation processes, the conditions such as temperature, updrafts, etc. do.

Reference to Figure 2 at the beginning of Section 2.1 for examples of the flight plans.

Page 5, line 18: What synoptic charts? Either provide a reference (paper or website) or synoptic maps as a figure. More information would also support the statements on page 5, lines 24-25.

Page 7, line 21: This would be a good place to discuss the variability in the winds and why, say, for 18 Jul the coastal and marine sections were relatively polluted (relatively stagnant winds unlike other days where faster winds introduced marine-soured air to the coast). In general, it would be useful to directly link relationships between winds, chemistry, and size. This is done to some extent, but should be clearly highlighted.

Page 7, lines 29-30: For the non-AMS crowd, please provide information on what these fragments are and clearly highlight which indicated a more oxidized OA.

Page 11, line 32: Is the "polluted case" Jul 18? Please define.

Page 14, line 12: This is true, for mineral dust compared to purely biological particles. However, we do not yet know the extent to which biological material within or on dust contributed to the nucleation of ice, i.e., determining if the mineral or biological components are what is nucleating on a single dust particle.

Page 15, lines 18-24: What is this value compared to other days the WIBS was operational? I understand the authors was to use the data to extrapolate to what might have been observed at aircraft level and solely focus on the flight days, but this does not seem valid due to the fact that information regarding the time and height of the aircraft over the site is not provided (i.e., to demonstrate what was observed on the ground was potentially observed aloft and 2). It would be helpful if the authors could provide more information to justify the use of the ground-based fluorescence to compare to aircraft, otherwise the INP concentrations compared to those estimated from parameterizations used for the aircraft data do not seem comparable. Also, the WIBS is briefly discussed, yet what types were used to calculate the concentrations (Type ABC)? Is this information found in a different publication on COPE? If not, please provide more details on the measurement.

Figure 2: The black arrows for "other" are distracting. It would be useful to remove these arrows from the picture as they are unneeded information.

Figure 4: Can the authors adjust the y axes in panels a, c, d, e, and f to show the same range? Panel b has much higher concentrations thus can remain.

Figure 7: Add that the dashed and solid lines are SMPS and PCASP distributions, respectively.

Figure 8: Jul 05 should be Jul 18 (typo in figure). Also, adding values for the vertical updraft speeds and total CN since these are discussed in the text.

---

## Author Comment (AC1) · 1 Jun 2016

Note: For clarity we use bold font for the referees' comments, plain font for our responses and italics to quote any text in the manuscript.

**Anonymous Referee #1**

**The study investigates cloud condensation nuclei (CCN) and ice nucleating particle (INP) characteristics in the marine-terrestrial region over the southwest peninsula of the UK in 2013. A comprehensive data set of aerosol chemical and microphysical parameters was acquired during several flights. One of the main results includes specific CCN number concentrations at 0.1 and 0.9 % supersaturation during clean and polluted conditions. Another result is that significant knowledge gaps still exist in predicting INP concentrations as the comparison of INP concentrations with existing parameterizations shows. The paper is written clearly, and the research and data analyses have been carried out very carefully with great attention to details. This work is a worthwhile contribution to the body of literature on CCN and INP, especially with respect to the marine-terrestrial interface and the differences between natural and anthropogenic CCN and INP contributions. I recommend publication with minor revisions.**

We thank the referee for their comments, which we will answer individually.

**General comments:**

**The authors introduce the greater topic of convective cloud formation and linkages to flash floods on the southern UK. However, the actual results of the COPE study are not interpreted in relation to this theme. To further improve this work, I recommend adding a more direct description of how COPE relates to convective cloud formation in the introduction, and a respective paragraph in the discussions section.**

We have added 2 paragraphs to the end of the introduction section

*"The main aim of COPE was to improve quantitative precipitation forecasting in numerical weather prediction (NWP) models. Clark et al. (2016) describe state-of-the-art convection-permitting models, which are used operationally for rainfall forecasting both in the UK and elsewhere. The development and application of these NWP models requires the use of parameterisations for processes that would be too small in scale or too costly computationally to explicitly evaluate on an operational basis. A parameterisation may be used, for example, to calculate the autoconversion rate of cloud water to rain, rather than explicitly evaluating processes such as collision-coalescence, condensation and evaporation over sub-grid timescales.*

*The development of improved parameterisations, and investigating the relative importance of different processes, involves the use of more detailed research models. For example, Connolly et al. (2009) described the aerosol-cloud-precipitation interaction model (ACPIM), one of several research models which investigators will use as part of COPE to study different cloud-aerosol interactions and microphysical processes. ACPIM requires information on aerosol composition and size distribution, as well as environmental variables, to initialise its explicit cloud microphysics scheme. The purpose of this paper is to describe the boundary layer aerosol measured during the aircraft campaign in terms of concentration, size, and composition, and to determine the main sources of CCN and INPs, in order to inform modelling studies aiming to improve forecasts of convective precipitation in the region. Additional measurements of cloud microphysics (e.g. Taylor et al., 2016) may be used to evaluate the importance of different cloud microphysical processes, while the X-*

*band radar measurements provide a dataset to evaluate the accuracy of the distribution and rate of precipitation in a regional model."*

**As far as I am aware, ice nuclei are referred to ice nucleating particles (INP) in the current literature. I recommend changing this throughout the manuscript.**

We have changed IN to INP(s) in the manuscript

**Specific comments:**

**p. 1, l. 16: specify if the concentration are given for STP or ambient conditions.**

The text now says

*"…2 – 3 µg m⁻³ (corrected to standard m³ at 1013.25 hPa and 273.15 K), …"*

**l. 20: What was the supersaturation at cloud base?**

It is difficult to quantitatively estimate supersaturation in-cloud, for example water vapour probes do not provide sufficient accuracy in supersaturation conditions. We have added a sentence to Section 3.4.1

*"…reasonable based on previous estimates of SST in cumulus clouds. For example, Politovich and Cooper (1988) calculated SST up to ~0.3% in cloud passes with a high fraction of air from near cloud base."*

**l. 22: It is not clear what the authors mean by "Marine organic aerosol (OA) had a similar mass spectrum to sea spray". What type of spectra are you comparing: only OA spectra in marine air masses to only sea spray spectra? I would expect that the OA spectrum does not show fragments of sodium and chloride (and other salts) while the sea spray spectrum is dominated by salts. Some clarification is needed.**

This now reads *"similar mass spectrum to previous measurements of sea spray OA"*

**p. 2, l. 2: The region is not particularly vulnerable because of convective cloud formation and flash floods. The region is particularly vulnerable because there is human activity where flash floods occur. In other words, if nobody lived there, there would be no vulnerability. Rephrase.**

We have clarified that the region is *"prone to flash flooding"*

**p. 3, l. 14f: Include a reference or explanation for the quantification of the uncertainty.**

We have added a reference to the instrument manual

**l. 19: a reference is missing.**

This now reads *"the uncertainty associated with the flow calibration and counting efficiency of the particle counter is typically ~6% (Trembath, 2013, p189)."*

**Section 2.1 Aircraft Measurements: Information about the inlet system is missing. Did you use a pressure controlled inlet? If yes, for which instruments, if not how did you treat/correct the data?**

We have added the following two sentences to the instrumental section

*"The CCNC sampled behind a pressure-controlled inlet set to 650 hPa."*

*"The online inboard aerosol instrumentation (i.e. the AMS, SP2, SMPS and CCNC) sampled using Rosemount inlets, which have sample efficiencies of approximately unity for particles smaller than 600 nm in diameter when sampling marine aerosol (Trembath et al., 2012)."*

**p. 4, l. 6: which densities did you use to calculate the volume?**

The sentence now reads

*"The collection efficiency was calculated as recommended by Middlebrook et al. (2012), and the calculated aerosol volume, calculated with volume mixing using organic and inorganic densities of 1.27 g cm-3 and 1.77 g cm-3 respectively (Cross et al., 2007), showed qualitative agreement with the aerosol volume measured by the SMPS and PCASP."*

**p. 5, l. 4: As far as I am aware, the recommendation is to use the latest reference for HYSPLIT as noted on their webpage. Further regarding the back trajectory analysis, did you run individual trajectories only or ensembles for each release time? If you did not run ensembles, how can you guarantee that the single trajectory is representative and accurate enough? Some more information on how you performed the analysis and why is needed.**

We have changed the HYSPLIT reference to the latest Stein et al. (2015). Regarding ensembles of runs and the accuracy of individual trajectories, this is already addressed in the text

*"Trajectories were initiated every 60 seconds from the aircraft flight track during the boundary layer aerosol runs.[…..] Although the turbulent mixing in the boundary layer means the accuracy of individual trajectories is uncertain, examining the general trend provides information on the history of the airmass, and possible changes in its cloud nucleating potential, which is determined by synoptic-scale winds."*

**l. 19: I would not state "any long-range transported" aerosol was washed out, but rather "most". Air masses that have experienced precipitation are not completely free from aerosol.**

We have changed *"any"* to *"the majority of any"*

**p. 6, l. 24: Information is missing on how you determined nss-chloride. We know from the literature you cite and further references that the AMS does measure a fraction of sea salt. So the signal at the chloride fragment m/z must be partially originating from sea salt.**

After a further survey of the literature, it does appear that the AMS is sensitive to sea-salt chloride to some extent, but with a much reduced (and highly variable) detection efficiency compared to less refractory species. Previous attempts to quantify sea salt with the AMS has showed that a correction factor is needed to account for the instrument's poor detection efficiency, but this correction factor appears to vary wildly depending on how the instrument is setup. This is summarised well by Schmale et al. (2015)[1]

*"Zorn (2009) found a scaling factor for chloride to sea salt between 150 and 220 in the South Atlantic, based on a comparison between measurements of an AMS and a particle-into-liquid sampler (PILS; Zorn et al., 2008), as opposed to the method here applied based only on AMS NO−3 quantification which resulted in a scaling factor ranging between of 3.15±0.20*
* * *
[1] Schmale, J., Schneider, J., Nemitz, E., Tang, Y. S., Dragosits, U., Blackall, T. D., Trathan, P. N., Phillips, G. J., Sutton, M. and Braban, C. F.: Sub-Antarctic marine aerosol: dominant contributions from biogenic sources, Atmos. Chem. Phys., 13(17), 8669–8694, doi:10.5194/acp-13-8669-2013, 2013.

*and 3.97±0.14 across all experiments. Ovadnevaite et al. (2012) determined a scaling factor between the NaCl+ reported by the AMS and the actual sea salt contained in PM1 (particulate matter with a diameter equal to or smaller 1 µm) of 51, while our experiments yielded a factor 13."*

A more recent paper by Nuaaman et al. (2015)[2] proposed another method to quantify sea-salt and non-sea-salt chloride with the AMS, but this does not appear to have made it through peer-review.

We could have used one of the methods from literature to calculate concentrations of sea salt and/or sea-salt chloride, but in light of the above it would be disingenuous to claim any sort of accuracy on the results. We have therefore decided the best course of action would be to remove the chloride measurements from the paper, as they serve no useful purpose and may be an order of magnitude out from the actual chloride concentration. We have added the following text to Section 2.1

*"The AMS can also be used to report chloride (Chl⁻) concentrations, which would be likely to be influenced by sea salt in a marine environment. Previous studies have attempted to quantify sea salt using the AMS, but the scaling factor used to correct the data for the instrument's poor detection efficiency of sea salt was highly variable (Nuaaman et al., 2015; Ovadnevaite et al., 2012; Schmale et al., 2013; Zorn, 2009). As no calibrations for sea salt were carried out in the field, we are unable to estimate the scaling factor, and thus the Chl⁻ concentration, with any reasonable accuracy, and we therefore do not report Chl⁻ concentrations in this analysis. The $SO_4^{2-}$ measurement may have a contribution from sea-salt sulphate, but this is likely to be of the order of 1% (Schmale et al., 2013)."*

We have also removed the chloride contribution to the data on Figures 9 and 10 panels (b) and (c), but this had almost no effect on the trends and $r^2$ values, and no modification of the text was required.

**p. 10, l. 5f: Was there no coarse mode from sea salt that was measured? More complete information on the size distribution is needed here.**

In response to this comment, and one below from Referee #2, we have added the following to Section 3.3

*"Coarse-mode aerosols were also measured, but the supermicron distribution is not shown in Fig. 7 as the bin sizing errors and overlap between the PCASP and CDP made the features difficult to distinguish. The mean diameters of the Aitken, accumulation and coarse modes are listed in Table 1. The mean diameter of the nucleation modes was likely to be smaller than the SMPS can reliably measure."*

We have also added some information regarding the average size of the supermicron aerosols, which is discussed below in a response to Referee #2 (see reference to Table 4).

**l. 7: From what I see in the figure, I cannot agree with the statement that all distributions were open at the lower end, e.g. in panel f, the red curve is not open (also in others). This needs some more detailed discussion. In case I misinterpret the meaning of "open" a clearer description is needed.**
* * *
[2] Nuaaman, I., Li, S.-M., Hayden, K. L., Onasch, T. B., Massoli, P., Sueper, D., Worsnop, D. R., Bates, T. S., Quinn, P. K. and McLaren, R.: Separating refractory and non-refractory particulate chloride and estimating chloride depletion by aerosol mass spectrometry in a marine environment, Atmos. Chem. Phys. Discuss., 15(2), 2085–2118, doi:10.5194/acpd-15-2085-2015, 2015.

We have rephrased to the following

*"Many of the measured distributions did not drop to near zero in the lowest bin, suggesting nucleation mode aerosols were present smaller than the 20nm size of the smallest bin."*

**p. 13, l. 4-9: Is there any particular reason why you did not apply kappa-Köhler theory? It seems only logical to derive a kappa value from the AMS measurements to compare it with previously determined kappa values.**

In a marine environment this is actually quite non-trivial to do. The calculation would require numerous assumptions regarding the mixing state of the aerosol- are the OA, MSA and SO4 internally- or externally-mixed? Is the NO3 real (this would imply some acidically-neutralised particles externally-mixed amongs a population of acidic particles)? Are any of these species internally- or externally-mixed with sea salt (it seems likely that they are to some extent)? What is an appropriate kappa for MSA?

A systematic investigation of all of these questions would be a significant undertaking and would be a largely theoretical exercise without measurements of kappa derived from CCN spectra, which were not available from the aircraft measurements. We could calculate one or more values of kappa, but it would be disingenuous to claim any sort of accuracy on the result.

**l. 29: The first factor determining CCN concentrations is the CN number concentration rather than the size distribution.**

We have added a paragraph to Section 3.4.1 that clarifies how the size distribution is more important than the CN concentration in these case studies.

*"The CN and CCN concentrations showed no correlation, and the fraction of CN active as CCN varied from 2% to 50%. This fraction was largest on 18 July, when the average particle size was the largest of all the flights. On all flights, the CCN fraction was lower inland than over the NW coast, particularly in the SWW cases where it was limited to single-figure percentages even at 0.9% SST. As noted in the previous section, the majority of inland particles were smaller than 60 nm and therefore too small to act as CCN. In these cases, the aerosol size distributions were a more important factor than the CN concentrations for determining CCN concentrations."*

**Fig. 1: I suggest to use the color code for altitude information and to insert numbers to indicate the age of the air mass.**

We have colour coded the figure for altitude and added a line of the mean trajectory with markers every 24 hours, and accordingly added to the text in section 3.1, regarding the 25 and 29 July and 03 August

*"the trajectories show the airmass remained in the lower troposphere, below the likely source of frontal precipitation."*

**Technical comments:**

**p. 1, l. 28: The sentence doesn't make sense grammatically.**

We have removed the word "that" so it now makes sense

**p. 3, l. 8:. "." Is missing after "al", also in some other occasions**

Done

**p. 10, l. 3: There is a grammatical error in the sentence.**

Changed to *"The standard deviations are plotted…"*

**Table 1: Include information on the year.**

Done

**Fig. 7: include information on what the solid and dashed lines represent in the captions.**

Done

**Fig. 8: A legend explaining what the rectangles and triangles are would help the reader to understand the message more easily and quickly.**

Done

**Fig. 9: The small graphs in each panel are not explained.**

We have added to the caption

*"Data from the SWW case and anthropogenic pollution (18 July) cases are separated due to the different meteorological conditions and greatly differing CCN concentrations."*

Also we have added the marine data for 18 July, which was previous missing. No modification of the text was necessary.

**Anonymous Referee #2**

**Taylor and coauthors provide a noteworthy and comprehensive set of aircraft observations of aerosols, specifically those which serve as cloud condensation nuclei (CCN), over variable source regions during the COPE field campaign. Additionally, they evaluate air mass sources and predict ice nucleating particle (INP) concentrations based on a set of different ice nucleation parameterization models. Although this work represents a detailed account of observations necessary to improve climate model simulations, there are a few issues that need to be resolved prior to publication in ACP.**

We thank the referee for their comments and suggestions.

**General comments:**

**In attempts to harmonize ice nucleation terminology, Gabor Vali and colleagues published a technical note to define terms used throughout the community. Please use ice nucleating particles (INPs) instead of IN, to align with Vali et al. (2015).**

**Vali, G., et al. Technical Note: A proposal for ice nucleation terminology. Atmospheric Chemistry and Physics 15.18 (2015): 10263-10270.**

As recommended by Vali et al. (2015), we have changed IN to INP(s) in the manuscript.

**Information regarding the instrumentation used is missing. First, what aircraft inlet was used? I am assuming from a brief statement later on in the text regarding not being able to sample in-cloud that an isokinetic inlet was used, but please provide the details in the methods section.**

We have added to the measurement section

*"The online inboard aerosol instrumentation (i.e. the AMS, SP2, SMPS and CCNC) sampled using Rosemount inlets, which have sample efficiencies of approximately unity for particles smaller than 600 nm in diameter when sampling marine aerosol (Trembath et al., 2012)."*

**Also, what instruments were used to measure CO, vertical velocity, and ice concentration? What are the units for these? Observations of CO and vertical velocity are presented in the manuscript, but information on the instrumentation is not provided. This would be alleviated by providing a few quick sentences in the methods, perhaps in a supporting measurements paragraph.**

We have added to the measurement section

*"The CO concentration was measured using an AeroLaser VUV fluorescence monitor model AL5002 (Gerbig et al., 1999), and the 3-D wind vector was measured with a de-iced Aventech aircraft-integrated meteorological measurement system (AIMMS)-20 turbulence probe (Beswick et al., 2008). A Stratton Park Engineering Company SPEC 2DS stereo probe was used to derive ice concentration, using the data processing rules described by Taylor et al. (2016)."*

**The altitude of the measurements and flight duration are vital pieces of information that should be provided. The authors do state the flights occurred below ~500 m, but measurements nearest to the surface could vary significantly from 500 m, depending on stratification, emissions, winds, etc. Even if provided by Leon et al. (2015), that information should be provided again here. This would also provide vertical context to the air mass trajectories as trajectories ending at 100 versus 500 m could lead to disparate sources. The authors could address this by providing an image of the**

**vertical sampling statistics, or even a range if sampling was only conducted at 500 m (this is not currently clear).**

We have added the altitude and duration of the aerosol runs to Table 1. We have also added some trajectories from the ground site from 03 August to Figure 1, which followed very similar paths to those released from the aircraft flight path. We have added to Section 2.1

*"The altitude and duration of the aerosol runs are listed in Table 1. Vertical profiles of potential temperature showed the boundary layer height was ~ 750 – 1250 m above mean sea level (AMSL; unless otherwise stated, all altitudes henceforth are AMSL), meaning these aerosol runs were conducted in the boundary layer."*

For transparency, here are the data used to estimate boundary layer height. There were not many dedicated profiles, so some of the traces are a little messy. Exeter airport is within the sampling region, but the aircraft was not based at this airport for the whole project, so there weren't landing profiles for every flight.

[Figure]

**Along these lines, on page 5, lines 19-20, the authors suggest that aerosols from long-range transport were likely removed via precipitation, but that depends on how high the trajectories were back in time. For instance, the height of a trajectory endpoint 3-4 days back could be thousands of meters. This statement would be more valid if vertical profiles of the trajectories were shown, perhaps as separate panel(s) in Figure 1. Also, what about trajectories at the ground site, since the authors present data from this location? A connection between the ground and location of aircraft observations would result from also running trajectories where the WIBS measurements were acquired.**

We have coloured the trajectories by altitude, and modified the text in section 3.1 so it now states (regarding the 25 and 29 July and 03 August)

*"This precipitation is likely to have washed out any the majority of any long-range transported aerosol as the trajectories show the airmass remained in the lower troposphere, below the likely source of frontal precipitation."*

We have added the run altitude and time lengths to Table 1. We have also added 2 trajectories to Figure 1 from the ground site released at the same time as the aerosol runs for 03 August, and added the following text to Section 3.1

*"Two additional trajectories were released from the ground site (at an altitude of 10m above ground level) on 03 August at 1100 and 1200UTC, the same time period as the aerosol runs, as the ground site measurements are used on this date in Sect. 3.5. These two ground-site trajectories follow a similar path to the ones released from the aircraft flight track on 03 August, meaning the aircraft and ground site were sampling from the same airmass on this date."*

Also please note our previous comment that establishes the aerosol runs were conducted in the boundary layer, and the text already states

*"Although the turbulent mixing in the boundary layer means the accuracy of individual trajectories is uncertain, examining the general trend provides information on the history of the airmass, and possible changes in its cloud nucleating potential, which is determined by synoptic-scale winds."*

**The NGA and NUGA data and discussion are not pertinent to the main themes of the manuscript.**

The concentrations of GA and UGA are useful for initiating model simulations of these and similar clouds. Indeed, this work is being carried out using COPE data as part of one of the co-authors' PhD. We have clarified this in the manuscript

*"It is also of interest to examine the concentrations of giant (dry diameter, $1 \leq D_p \leq 10 \ \mu m$) and ultragiant particles ($D_p > 10 \ \mu m$), as these may have an enhanced effect on precipitation formation via the warm rain process (Johnson, 1982), and our observations may be used to inform modelling studies (e.g. Blyth et al., 2013)"*

**Further, the values are quite low; are these values of any significance compared to total CCN concentrations?**

The text already states

*"In all cases, $N_{GA}$ and $N_{UGA}$ comprised a very small fraction of the total aerosol number concentration, including the concentration of particles large enough to affect CCN populations."*

**I suggest eliminating this information as it is only discussed in one small paragraph and does not add any significance to the main conclusions. What the authors could instead do is provide mean size for each day, perhaps for each size distribution mode, to provide some sizing context to the CCN.**

We have kept the section in for the reasons given above, but have also added in Table 4 which lists the mean diameters of each aerosol mode

| Date | Mean diameter (µm) | | |
|---|---|---|---|
| | Aitken mode | Accumulation mode | Coarse mode |
| **NW Coast** | | | |
| 05 July | 0.048 | 0.188 | 3.92 |
| 18 July | N/A[a] | 0.125 | 3.25 |
| 25 July | 0.060 | 0.184 | 4.18 |
| 28 July | 0.049 | 0.153 | 3.04 |

| | | | |
|---|---|---|---|
| 29 July | 0.041 | 0.155 | 4.20 |
| 03 August | 0.043 | 0.158 | 4.00 |
| | **Inland** | | |
| 05 July | N/A[a] | N/A[a] | 3.71 |
| 18 July | N/A[a] | 0.114 | 3.13 |
| 25 July | 0.055 | 0.169 | 3.76 |
| 28 July | 0.051 | 0.145 | 3.56 |
| 29 July | N/A[a] | 0.151 | 3.61 |
| 03 August | 0.057 | 0.155 | 3.87 |

[a]Mode not distinct from surrounding distribution

Table 1 - Mean diameter of distinct aerosol modes measured over the NW coast and inland on the different cast study flights.

**Are there enough sampling statistics to look at vertical profiles of the aerosol measurements? That can provide quite a bit of insight into the sources and transport of aerosols. At the very least, this can be done with the SMPS, PCASP, and CDP. Also, RH, mixing ratio, temperature profiles would be helpful to show.**

On most flights there were no dedicated profiles, just a single boundary layer run followed by cloud sampling. This is a (modified) part of Fig. 4 from Taylor et al. (Taylor et al., 2016)[3], showing the time series of sampling altitude from 03 August, where "Line AB" and "Line CD" were sampling different lines of cloud.

[Figure]

Profile information was, therefore, obtained from out-of-cloud data during the cloud sampling. The vertical profile of aerosol consistently showed that there was a marked drop in aerosol concentration above the boundary layer compared to within the boundary layer. The vertical sampling statistics prevented any more detailed conclusions about the aerosol profiles

**Why is SEM-EDX introduced in the methods, but no results are provided? The single particle chemistry would serve as quite useful information for characterizing the aerosol.**

We have modified the text of section 2.1 so it now reads

*"An example filter composition measurement from the COPE campaign is presented by Leon et al. (Leon et al., 2015). In our analysis, we only use the filter measurements to estimate the concentration of mineral dust for use in INP parameterisations."*

And Section 3.5 now reads
* * *
[3] Taylor, J. W., Choularton, T. W., Blyth, A. M., Liu, Z., Bower, K. N., Crosier, J., Gallagher, M. W., Williams, P. I., Dorsey, J. R., Flynn, M. J., Bennett, L. J., Huang, Y., French, J., Korolev, A. and Brown, P. R. A.: Observations of cloud microphysics and ice formation during COPE, Atmos. Chem. Phys., 16(2), 799–826, doi:10.5194/acp-16-799-2016, 2016.

*"Mineral dust concentration and size distribution were measured using the BAe-146 filter system, by combining the 'silicates' and 'mixed silicate' categories described by Young et al. (2016). Leon et al. (2015) show a more detailed breakdown of the composition of particles from 03 August."*

As an additional note, the bulk of the paper is focused on submicron aerosol, particularly the accumulation mode, but as we have noted in Section 3.5

*"In a previous study, the concentrations of particles collected on the filters agreed well with in-situ probes for particles larger than 0.5 µm, but the concentrations at smaller particle sizes compared poorly (Young et al., 2016)."*

**Given the temperature range presented in the last section, I would expect the ice particle number concentration to equate to several orders of magnitude higher than INP concentrations (i.e., Hallett-Mossop). The authors should revise this section to include secondary ice formation as a plausible reason and tone down the element of surprise.**

The text already states

*"The measured ice concentrations are from a series of penetrations through a cloud in the early stages of development, when secondary ice processes were thought to have at most a minor influence on ice concentrations* (Taylor et al., 2016)*."*

**Specific comments:**

**The abstract could use a sentence or two regarding the broader impacts and motivation for the work. By adding some "big picture" material, the significance of this work is evident right off the bat.**

We have added to the start of the abstract

*"Heavy rainfall from convective clouds can lead to devastating flash flooding, and observations of aerosols and clouds are required to improve cloud parameterisations used in precipitation forecasts"*

**Introduction: Similar to the abstract, end the first paragraph with a direct statement to segue into the next paragraph, i.e., something along the lines of, "The potential of flooding from persistent convective clouds along the peninsula demonstrate the importance to understand cloud formation in this region."**

We have added

*"The potential of convective clouds to generate persistent, heavy rainfall, and consequent flooding, along the peninsula demonstrates the importance of understanding cloud formation and development in the region."*

**As I and potentially other readers are not familiar with this region, it would be helpful to point out Figures 1 and 2 in the first paragraph for geographical context.**

The first paragraph of the introduction now says *"The southwest peninsula of the UK (shown in Figs. 1 and 2)"*

**Page 1, line 27: Define that the parameterizations are for prediction of ice nucleating particle concentrations. Parameterizations is somewhat vague.**

This now reads

*"Sources of ice-nucleating particles (INPs) were assessed by comparing different parameterisations used to predict INP concentrations, using measured aerosol concentrations as input."*

**Page 2, line 13: Can also inhibit cold precipitation by reducing riming efficiency of descending ice particles in a mixed-phase cloud system.**

We have added *"which also affects riming efficiency in mixed-phase clouds (Klett and Davis, 1973)"*

**Also, replace "lower" with "subzero".**

Done

**Page 2, line 14: "..such as riming and the Hallett-Mossop rime-splintering processes."**

We have added "such as the Hallett-Mossop rime-splintering processes" as riming in itself is not a secondary ice formation process.

**Also, the aerosols themselves do not "initiate" secondary ice formation processes, the conditions such as temperature, updrafts, etc. do.**

*"Initiate"* is now changed to *"lead to"*

**Reference to Figure 2 at the beginning of Section 2.1 for examples of the flight plans.**

The first paragraph now has the sentence

*"The flight plans of the boundary-layer aerosol runs in each flight are shown in **Error! Reference source not found.**, and the altitude and duration of the aerosol runs are listed in Table 1"*

**Page 5, line 18: What synoptic charts? Either provide a reference (paper or website) or synoptic maps as a figure. More information would also support the statements on page 5, lines 24-25.**

We have clarified that the charts are *"UK Met Office synoptic charts (available at http://www1.wetter3.de/error_adblocker_archiv_ukmet.html)"*

**Page 7, line 21: This would be a good place to discuss the variability in the winds and why, say, for 18 Jul the coastal and marine sections were relatively polluted (relatively stagnant winds unlike other days where faster winds introduced marine-soured air to the coast). In general, it would be useful to directly link relationships between winds, chemistry, and size. This is done to some extent, but should be clearly highlighted.**

The end of Section 3.2.2 now reads

*"…this suggests that local fossil fuel emissions were at least partly responsible for the increased OA inland on 05 July. Some of the easternmost trajectories passed over the southern tips of Wales and Ireland, which are an alternative source of the increased anthropogenic influence inland on 05 July.*

*"18 July was somewhat of an outlier in terms of the COPE case studies in that the aerosol mass loadings were much higher than the other flying days. The back trajectories showed that the airmass had come from England, Wales and/or Ireland, so a strong anthropogenic influence would be expected, in contrast to the clean marine air from the southwest seen in the other cases. There was not much difference between the aerosol measured inland and*

*on the NW coast, as the air over both areas had come from the East, bringing anthropogenic pollution to the region."*

In Section 3.3 we have also added about 18 and 05 July

*"The features of the marine aerosol size distribution seen on other days, such as a prominent Aiken mode and Hoppel dip, were also absent on this day due to the lack of a marine influence. The number concentration of aerosols in the accumulation mode on 18 July was significantly higher than on any of the other days studied, due to the anthropogenic pollution influence. The concentration of particles in the inland nucleation mode on 05 July was also higher than on any of the other flights, and this mode was broader than the other flights, which may be due to the stagnant winds allowing photochemical processing to occur within the airmass with minimal mixing of clean marine air."*

**Page 7, lines 29-30: For the non-AMS crowd, please provide information on what these fragments are and clearly highlight which indicated a more oxidized OA.**

This now says

*"The spectrum is dominated by peaks at m/z 28 and 44 from $CO^+$ and $CO_2^+$, which are prescribed to be equal in the default fragmentation table. As is typical in oxidised OA spectra, these peaks are much larger than m/z 43 from $C_3H_7^+$ or $H_3C_2O^+$, as the $CO^+$ and $CO_2^+$ ions are more likely to be formed by fragmentation of more oxidised organic molecules."*

**Page 11, line 32: Is the "polluted case" Jul 18? Please define.**

Done

**Page 14, line 12: This is true, for mineral dust compared to purely biological particles. However, we do not yet know the extent to which biological material within or on dust contributed to the nucleation of ice, i.e., determining if the mineral or biological components are what is nucleating on a single dust particle.**

We have added a note about internal mixing

*"Their concentrations at cloud formation levels are generally much lower than mineral dusts, though there is evidence that ice-active biological material internally-mixed with mineral dust and may enhance the ice nucleating potential of the dust (Augustin-Bauditz et al., 2016)"*[4]

**Page 15, lines 18-24: What is this value compared to other days the WIBS was operational? I understand the authors was to use the data to extrapolate to what might have been observed at aircraft level and solely focus on the flight days, but this does not seem valid due to the fact that information regarding the time and height of the aircraft over the site is not provided (i.e., to demonstrate what was observed on the ground was potentially observed aloft and 2). It would be helpful if the authors could provide more information to justify the use of the ground-based fluorescence to compare to aircraft, otherwise the INP concentrations compared to those estimated from parameterizations used for the aircraft data do not seem comparable.**

Please see our earlier responses regarding the trajectories released from the ground site and turbulent mixing in the boundary layer. We have also added this to Section 3.5
* * *
[4] Augustin-Bauditz, S., Wex, H., Denjean, C., Hartmann, S., Schneider, J., Schmidt, S., Ebert, M. and Stratmann, F.: Laboratory-generated mixtures of mineral dust particles with biological substances: characterization of the particle mixing state and immersion freezing behavior, Atmos. Chem. Phys., 16(9), 5531–5543, doi:10.5194/acp-16-5531-2016, 2016.

*"There may be some difference between the PBAP concentration measured at the ground site that that at the altitude of the aircraft measurements. However, it is unlikely that the PBAP concentration at ~500 m would be three orders of magnitude higher than at ground level, which is what would be required for the INP concentrations calculated using the T13-F parameterisation to agree with the measured ice concentrations."*

In other words, you might expect more PBAP on the ground, but even when using the ground-based measurements the PBAP INP concentrations are still much lower than the ice measurements.

**Also, the WIBS is briefly discussed, yet what types were used to calculate the concentrations (Type ABC)? Is this information found in a different publication on COPE? If not, please provide more details on the measurement.**

The text already states

*"Healy et al. (2014) made concurrent measurements using a UV-APS and a WIBS, and found the WIBS channel 3 (which uses the same fluorophore as the UV-APS) was well correlated with the UV-APS fluorescent concentration, but a factor of 2.7 higher due to the instruments' different size ranges. […] The fluorescent concentration measured using channel 3 was $50 \pm 26$ L$^{-1}$, which corresponds to a UV-APS equivalent concentration of $19 \pm 10$ L$^{-1}$. Using this concentration in the T13-F parameterisation generates INP concentrations…"*

Using the ABC scheme from Perring et al. (2015)[5] that the reviewer is referring to, the FL3 concentration is equivalent to the sum (C + AC + BC + ABC), but that seems like a more complicated way of saying the same thing.

**Figure 2: The black arrows for "other" are distracting. It would be useful to remove these arrows from the picture as they are unneeded information.**

Done

**Figure 4: Can the authors adjust the y axes in panels a, c, d, e, and f to show the same range? Panel b has much higher concentrations thus can remain.**

Done, for figures 3 and 4

**Figure 7: Add that the dashed and solid lines are SMPS and PCASP distributions, respectively.**

Done

**Figure 8: Jul 05 should be Jul 18 (typo in figure). Also, adding values for the vertical updraft speeds and total CN since these are discussed in the text.**

We have fixed the error in the axis, and added the CN concentration and vertical wind distributions to the graph. We have also made reference to these in the first paragraph of Section 3.4.1, and also added a second paragraph comparing the CN and CCN concentrations:

*"The CN and CCN concentrations showed no correlation, and the fraction of CN active as CCN varied from 2% to 50%. This fraction was largest on 18 July, when the average particle size was the largest of all the flights. On all flights, the CCN fraction was lower inland than*
* * *
[5] *Perring, A. E., et al. (2015), Airborne observations of regional variation in fluorescent aerosol across the United States, J. Geophys. Res. Atmos., 120, 1153–1170, doi:10.1002/2014JD022495.*

*over the NW coast, particularly in the SWW cases where it was limited to single-figure percentages even at 0.9% SST. As noted in the previous section, the majority of inland particles were smaller than 60 nm and therefore too small to act as CCN. In these cases, the aerosol size distributions were a more important factor than the CN concentrations for determining CCN concentrations."*

The referees and editor should also note that we have updated the traces for DeMott et al. (2015) and Niemand et al. (2012) in Figure 11. The data processing method underwent a minor change in response to referees' comments that Young et al. (2016)[6] received. The plot and relevant numbers in the text are now consistent with the final (ACP) version of Young et al. (2016). The traces moved slightly but this did not affect the conclusions or require any changes to the text.
* * *
[6] Young, G., Jones, H. M., Darbyshire, E., Baustian, K. J., McQuaid, J. B., Bower, K. N., Connolly, P. J., Gallagher, M. W. and Choularton, T. W.: Size-segregated compositional analysis of aerosol particles collected in the European Arctic during the ACCACIA campaign, Atmos. Chem. Phys., 16(6), 4063–4079, doi:10.5194/acp-16-4063-2016, 2016.

---

## Author Response (AR2)

Suggestions for revision or reasons for rejection (will be published if the paper is accepted for final publication)

Taylor et al. present a thorough and comprehensive analysis of measurements from the COPE field campaign. The large dataset enables the authors to draw several conclusions regarding the coastal versus inland and flight-to-flight variability of aerosol in terms of chemistry, size, and cloud nucleating abilities. Although the manuscript has improved from its original version, I find it somewhat difficult to follow due to the level of density of the measurements. It is a difficult task to eloquently discuss such a vast set of observations, but with some restructuring, the story would flow much better. The science is sound and the data needed are already presented, but I recommend some minor revision of the structure prior to publication in its final form.

We thank the referee for their useful suggestions and comments.

I suggest the authors consider limiting the first several sections following the methods to simply and concisely stating the results, followed by a discussion section linking the different yet complementary measurements for each flight, then how and why these vary between the flights. As it stands, the discussion jumps back and forth between different flights and measurements, which renders it difficult to follow. The synergy between the different measurements is unique and interesting, for instance the authors do this for the correlation plots between CCN and various aerosol observations (Fig 9). The discussion corresponding to this figure is convincing and a great example of what could be conducted for all the analyses to make the combination of the measurements stronger. As an example, the authors could directly explain how the size distributions can be explained by the chemistry and meteorology. Analysis such as those presented in Fig 9 are the heart of the paper and demonstrate the dependence of aerosol climate effects on aerosol properties.

If this restructuring is done, I would also suggest reordering so the results for size are provided prior to the chemistry, then, as the authors currently have, finish with the aerosol climate related measurements/parameterizations.

We have split the old results and discussion section into two separate sections. Some of the more analysis-type text from the old section has been moved into the new section 4.1 which contains analysis about the links between meteorology, chemistry and aerosol properties. The results section is now more results-focused. Additionally we have moved the size distribution results before the chemical analysis as the reviewer suggested. We stopped short of dividing the results up into a separate description of each case study as many of the cases had similar features, so it was more useful to describe and analyse them together rather than make several similar descriptions which might run the risk of boring the reader.

**This would also alleviate issues with discussing results before they are introduced, e.g., on page 12, lines 22 and 32 (CCN and PBAP not yet shown) and page 13, line 5 (CCN again).**

We have removed the parts previously on P12L22 and P13L5 in the restructuring. The part on P12L32 is a generic statement referring to PBAP in general, rather than our measurements of PBAP, so didn't need changing.

**General comments:**

The authors state that Rosemount inlets were used for the internal instrumentation. What are the specs on these inlets? Was there not a main inlet to the aircraft (i.e., isokinetic or CVI)? If a non-CVI inlet was used, how did the authors filter out interference from very small cloud droplets in the overlap region of the PCASP and CDP size ranges? This at least should not be an issue for the

**CDP considering the particles need a relatively substantial amount of water for the CDP to detect them.**

The specifications of the Rosemount inlets are listed in Trembath et al. (2012)1, which is referenced in Section 2.1 when discussing the inlet. Regarding the use of other inlets, the text makes clear *"The online inboard aerosol instrumentation (i.e. the AMS, SP2, SMPS and CCNC) sampled using Rosemount inlets"* i.e. there were no other inlets. Regarding small clouds drops, we have clarified in Section 2.1 *"A typical flight plan involved performing below-cloud aerosol runs in the boundary layer before making measurements of clouds at higher levels."* i.e. we were not sampling near clouds during the aerosol runs. Additionally, the text already states that the PCASP and CDP were wingmounted.

**Could the authors conclude that 05 and 18 Jul were more polluted based on the fact they had the most stagnant winds (i.e., less clean out and more opportunity to enable gas phase partitioning/particle aging)?**

The manuscript already contained the following text, though it has now moved to the new section 4.1.1:

"The concentration of particles in the inland nucleation mode on 05 July was also higher than on any of the other flights, and this mode was broader than the other flights, which may be due to the stagnant winds allowing photochemical processing to occur within the airmass with minimal mixing of clean marine air."

And we have now added in just afterwards

"However, most of the pollution is likely to be advected from urban areas further east, rather than emitted locally and allowed to build up, as the southwest peninsula of the UK is not a large source region of urban pollution."

Although the WIBS observations are interesting, because they are only discussed briefly, they seem superfluous in the context of the aircraft measurements. How representative are the ground-based observations for what is aloft? Comparing ground-based aerosol and aircraft-based cloud properties is somewhat of a stretch, unless the authors clearly state this potential source for inconsistency. Further, the entire focus of the paper is on the aircraft measurements. If the authors choose to keep the WIBS measurements, there are other ground-based aerosol observations that could be used to support their findings from COPE (Leon et al. 2015). More of an emphasis on ground-based observations would balance the story nicely. However, this may also increase the length of the paper and hinder the focus.

The reason we included the WIBS measurements from the ground site is we wanted an estimate of INP from primary biological sources and we did not have a WIBS or similar instrument on the aircraft. We have already included a comment on comparing ground- and aircraft-based measurements, which states

"There may be some difference between the PBAP concentration measured at the ground site to that at the altitude of the aircraft measurements. However, it is unlikely that the PBAP concentration at ~500 m would be three orders of magnitude higher than at ground level, which is what would be required for the INP concentrations calculated using the T13-F parameterisation to agree with the measured ice concentrations."

We agree with the referee that adding more detail from the ground site would make the paper longer and hinder the focus. Also, having looked at the other ground-site data, we did not feel it added anything useful to the analysis presented in this paper. We have added to Section 2.2 so it now reads

<sup>1 Trembath, J., Bart, M. and Brooke, J.: Efficiencies of Modified Rosemount Housings for sampling Aerosol on a Fast Atmospheric Research Aircraft., FAAM Tech. Note [online] Available from:

http://www.faam.ac.uk/index.php/component/docman/doc\_download/1673-inlet-efficiency (Accessed 24 July 2014), 2012.

"The aerosol properties measured at the ground site followed the same trends as those measured aboard the BAe-146, though it is not in the scope of this paper to provide an in-depth comparison. In this analysis, we only use the size-resolved measurements of fluorescent aerosol made by a University of Hertfordshire wideband integrated bioaerosol sensor (WIBS) 4M, which we consider to be primary biological aerosol particles (PBAP), as this measurement was not available on the BAe-146."

**Specific comments:**

**Page 2, line 1: Lower than what?**

This now reads "lower than the measured first ice concentrations"

Page 4, line 17: Briefly elaborate on "active drying".

This now reads "The PCASP inlet has a dried sheath flow..."

Page 4, line 23: The chloride ion is traditionally designated as Cl-. Fixed

**Section 2.2: It is stated later in the text that the description of the WIBS is provided, yet not a whole lot of details on the WIBS here.**

We have added some further description of the WIBS to section 2.2

"...a University of Hertfordshire wideband integrated bioaerosol sensor (WIBS) 4M, as this measurement was not available on the BAe-146. The WIBS was was mounted atop the ~8m-high sampling tower. The particle size distribution in the range  $0.4 - 12 \,\mu$ m (optical equivalent diameter) is measured by light scattering at a wavelength of 635nm. When a scattering particle is detected, two xenon lamps fire sequential UV pulses at at 280 and 370 nm, which can cause fluorescent emissions that are detected in three channels measuring the ranges 310–400 nm (channel F1), 420–650 nm (channel F2), and 420–650 nm (channel F3) (Healy et al., 2012)."

**Page 8, lines 6-10: Provide a sentence or two on why it is important to look at sulfate neutralization.**

We have added the following to what is now section 3.3.1

"The aerosol acidity, investigated here in terms of sulphate neutralisation, influences gas-particle partitioning. For example, nitrate aerosol cannot stably coexist when internally-mixed with particle-phase sulphuric acid, but may do so with ammonium sulphate."

Page 11, line 14: How exactly was this done? SMPS and PCASP measure electric mobility and optical diameters, respectively. Shouldn't a conversion be done in this case to permit combining the distributions? Along these lines, it is difficult to see the dashed versus solid lines for the respective instruments in the figure.

We have moved this to the experimental section and added the following

"The SMPS data were normalised to the PCASP by scaling the SMPS concentrations in each bin so the concentrations in the overlap region matched those of the PCASP. As is shown in later analysis, the particles are composed predominantly of secondary aerosol and are likely to be quasi-spherical, meaning any shape correction for the SMPS distributions would be minimal. The difference in the size of the PCASP bins derived using refractive index of 1.5 compared to using values 1.53 for ammonium sulphate and 1.44 for sulphuric acid is ~5%, which is equivalent to half a bin. Taking this difference into account when normalising the SMPS distributions gave a 3 - 6% uncertainty in the normalised SMPS distributions."

**Section 3.3: At times, it is difficult to ascertain if the authors are talking about the coastal, inland, or both size distributions. Please clarify throughout when drawing conclusions.**

Some of the confusion is probably due to an error in the first sentence, which introduced the size distributions as being over the NW coast, when in fact only half of them were. We have fixed the error, and clarified the ambiguities in the new section 3.2 and 4.1.

Section 3.4.2: This section also contains sizing observation; the section title is not entirely inclusive. This would be alleviated if the R&D section was restructured and separated. The section title is now "Aerosol chemistry, CN and CCN"

**Fig 10: This figure is somewhat distracting and draws away from the more exciting results. The authors could eliminate this figure and simply state that none of the parameters in Fig 9 correlated well at 0.9%.**

We have removed the old Fig. 10 and the discussion of it in the text, and replaced it with "We also investigated correlations between the x-variables in Fig. 9 and the CCN concentrations at 0.9% SST, but the correlations were not as good. This is likely to be because smaller particles activate at 0.9% SST, which are less associated with particle mass and may be smaller than the PCASP's detection range."

Fig 11: How were lower ice concentrations observed during updrafts than downdrafts? Sullivan et al. (2016) demonstrated how vertical updrafts are a key element to cloud ice formation (can explain up to 48% and 89% of ice crystal number). Also, I am assuming these are number concentrations, and if so, relabel the axis to calculated INP or ice crystal concentration. The Sullivan et al. (2016) study was looking at cirrus clouds on a global scale, whereas the observations described by Taylor et al. (2016)2 were of the order km to hundreds of m in short-lived convective clouds in in the troposphere, where secondary ice plays an important role. This is discussed in detail in Taylor et al.

We have modified the axis label on the figure.

Sullivan, S.C., Lee, D., Oreopoulos, L., and Nenes, A (2016) The role of updraft velocity in temporal variability of cloud hydrometeor number, Proc. Nat. Acad. Sci., in press.

<sup>2 Taylor, J. W., Choularton, T. W., Blyth, A. M., Liu, Z., Bower, K. N., Crosier, J., Gallagher, M. W., Williams, P. I., Dorsey, J. R., Flynn, M. J., Bennett, L. J., Huang, Y., French, J., Korolev, A. and Brown, P. R. A.: Observations of cloud microphysics and ice formation during COPE, Atmos. Chem. Phys., 16(2), 799–826, doi:10.5194/acp-16-799-2016, 2016.

**Aerosol measurements during COPE: composition, size and sources of CCN and INPs at the interface between marine and terrestrial influences**

J. W. Taylor1, T. W. Choularton1, A. M. Blyth2, M. J. Flynn1, P. I. Williams1,3, G. Young1, K. N. Bower1, J. Crosier1,3, M. W. Gallagher1, J. R. Dorsey1,3, Z. Liu1, P. D. Rosenberg4

[revised manuscript text omitted]

|              | Flight | Run             | Run             | SO 4 (µ | gOrg                  | MSA                   | NO 3       | NH 4       | BC                    | CCN 0.1%                | CCN                             | CN                    | N GA             | N UGA   |
|--------------|--------|-----------------|-----------------|--------------------|-----------------------|-----------------------|-----------------------|-----------------------|-----------------------|-------------------------|---------------------------------|-----------------------|-----------------------------|--------------------|
| Date         | number | altitude
(m) | length
(min) | m -3 )  | (µg m -3 ) | (µg m -3 ) | (µg m -3 ) | (µg m -3 ) | (ng m -3 ) | SST (cm -3 ) | Column 2
(cm -3 ) | 2 (cm -3 ) | ( cm -1 ) | (L -1 ) |
| NW Coast     |        |                 |                 |                    |                       |                       |                       |                       |                       |                         |                                 |                       |                             |                    |
| 05 July 2013 | 3 B786 | 330±140         | 12              | 2.19               | 1.27                  | 0.10                  | 0.11                  | 0.11                  | 12                    | -                       | -                               | -                     | 1.3                         | 3                  |
| 18 July 2013 | 3 B788 | 612±14          | 9               | 5.0                | 11.88                 | 0.32                  | 5.15                  | 3.55                  | 530                   | 1540±150                | 2860±110                        | 5030±220              | 0.71                        | 3                  |
| 25 July 2013 | 3 B789 | 388±16          | 9               | 2.34               | 0.55                  | 0.13                  | 0.06                  | 0.16                  | 10                    | 230±21                  | 500±100                         | 990±150               | 0.83                        | 3                  |
| 28 July 2013 | 3 B790 | 382±13          | 8               | 1.31               | 0.96                  | 0.04                  | 0.17                  | 0.48                  | 26                    | -                       | -                               | -                     | 1.4                         | 4                  |
| 29 July 2013 | 3 B791 | 396±22          | 8               | 1.48               | 0.52                  | 0.05                  | 0.07                  | 0.28                  | 18                    | 168±25                  | 440±140                         | 1400±400              | 1.6                         | 12                 |
| 03 Aug 2013  | 3 B792 | 405±23          | 8               | 1.73               | 0.32                  | 0.09                  | 0.07                  | 0.18                  | 7                     | 146±9                   | 450±120                         | -                     | 1.2                         | 6                  |
| Inland       |        |                 |                 |                    |                       |                       |                       |                       |                       |                         |                                 |                       |                             |                    |
| 05 July 2013 | 3 B786 | 540±80          | 64              | 1.56               | 2.62                  | 0.18                  | 0.71                  | 0.8                   | 97                    | -                       | -                               | -                     | 1.60                        | 4                  |
| 18 July 2013 | 3 B788 | 615±11          | 20              | 4.0                | 10.13                 | 0.28                  | 4.55                  | 3.28                  | 470                   | 1230±120                | 2740±160                        | 7200±800              | 0.75                        | 3                  |
| 25 July 2013 | 3 B789 | 570±70          | 39              | 1.1                | 0.67                  | 0.10                  | 0.17                  | 0.67                  | 32                    | 278±27                  | 500±50                          | 8000±6000             | 0.7                         | 3                  |
| 28 July 2013 | 3 B790 | 520±30          | 18              | 1.1                | 0.97                  | 0.04                  | 0.23                  | 0.5                   | 30                    | -                       | -                               | -                     | 0.9                         | 4                  |
| 29 July 2013 | 3 B791 | 530±40          | 20              | 1.4                | 0.54                  | 0.05                  | 0.1                   | 0.39                  | 31                    | 164±49                  | 410±120                         | 9000±5000             | 1.1                         | 4                  |
| 03 Aug 2013  | 3 B792 | 550±40          | 20              | 1.2                | 0.42                  | 0.08                  | 0.09                  | 0.49                  | 30                    | 212±33                  | 440±70                          | 7000±2000             | 0.7                         | 3                  |
|              |        |                 |                 |                    |                       |                       |                       |                       |                       |                         |                                 |                       |                             |                    |

|           | Mean diameter (µm) |                  |        |  |  |  |
|-----------|--------------------|------------------|--------|--|--|--|
| Date      | Aitken             | Accumulation     | Coarse |  |  |  |
|           | mode               | mode             | mode   |  |  |  |
|           | NW Coast           |                  |        |  |  |  |
| 05 July   | 0.048              | 0.188            | 3.92   |  |  |  |
| 18 July   | N/A a   | 0.125            | 3.25   |  |  |  |
| 25 July   | 0.060              | 0.184            | 4.18   |  |  |  |
| 28 July   | 0.049              | 0.153            | 3.04   |  |  |  |
| 29 July   | 0.041              | 0.155            | 4.20   |  |  |  |
| 03 August | 0.043              | 0.158            | 4.00   |  |  |  |
|           | Inland             |                  |        |  |  |  |
| 05 July   | N/A a   | N/A a | 3.71   |  |  |  |
| 18 July   | N/A a   | 0.114            | 3.13   |  |  |  |
| 25 July   | 0.055              | 0.169            | 3.76   |  |  |  |
| 28 July   | 0.051              | 0.145            | 3.56   |  |  |  |
| 29 July   | N/A a   | 0.151            | 3.61   |  |  |  |
| 03 August | 0.057              | 0.155            | 3.87   |  |  |  |

Table 2 - Mean diameter of distinct aerosol modes measured over the NW coast and inland on the different cast study flights.

aMode not distinct from surrounding distribution

|           |               | Uncentred R    |                |                 |            |            |
|-----------|---------------|----------------|----------------|-----------------|------------|------------|
| Date      | Sea spray     | Marine         | Organic factor | НОА             | OOA-1      | OOA-2      |
|           | (Ovadnevaite  | biogenic SOA   | (Chang et al., | (Morgan et al., | (Morgan et | (Morgan et |
|           | et al., 2011) | (Chang et al., | 2011)          | 2010)           | al., 2010) | al., 2010) |
|           |               | 2011)          |                |                 |            |            |
| 05 July   | 0.99          | 0.84           | 0.99           | 0.47            | 0.81       | 0.83       |
| 25 July   | 0.97          | 0.89           | 0.97           | 0.51            | 0.78       | 0.83       |
| 29 July   | 0.99          | 0.79           | 0.98           | 0.38            | 0.83       | 0.79       |
| 03 August | 0.95          | 0.85           | 0.96           | 0.46            | 0.79       | 0.79       |

Table 3 - Uncentred R (also known as normalised dot product) between marine OA measured on the SWW case studies and several literature spectra. m/z 28 was not used for the correlations due to gas-phase interference.

Table 4 - Comparison of the m/z 30/46 ratios measured in the AMS aboard the BAe-146.

| Date      | (m/z 30) / (m/z 46) ratio                     |          |        |  |  |  |
|-----------|-----------------------------------------------|----------|--------|--|--|--|
|           | (NH 4 )NO 3 calibration | NW Coast | Inland |  |  |  |
| 05 July   | 1.77                                          | 6.33     | 2.78   |  |  |  |
| 18 July   | 1.65                                          | 1.67     | 1.66   |  |  |  |
| 25 July   | 1.48                                          | 6.61     | 3.05   |  |  |  |
| 28 July   | 1.51                                          | 4.40     | 3.14   |  |  |  |
| 29 July   | 1.37                                          | 7.84     | 6.15   |  |  |  |
| 03 August | 1.33                                          | 8.68     | 4.01   |  |  |  |